# Quantum many-body dynamics on the star graph

Andrew Lucas

Department of Physics, Stanford University, Stanford, CA 94305, USA

ajlucas@stanford.edu                                      March 15, 2019

**Abstract:** We study 2-local Hamiltonian quantum systems, consisting of qubits interacting on the star graph of $N$ vertices. We numerically demonstrate that these models are generically non-integrable at infinite temperature, and find evidence for a finite temperature phase transition to a glassy phase in generic models. Operators can become complicated in constant time: we explicitly find that there is no bound on out-of-time-ordered correlators, even at finite temperature. Operator growth is not correctly modeled by stochastic quantum dynamics, including Brownian Hamiltonian dynamics or random unitary circuits. The star graph (and similar constructions) may serve as a useful testing ground for conjectures about universality, quantum chaos and Planckian dissipation in $k$-local systems, including in experimental quantum simulators.

## 1  Introduction

Understanding and ultimately controlling the dynamics of quantum information is a problem with suprisingly broad applications. One obvious application of such "technology" is the construction of a quantum computer [1]; less obvious are the profound relations that have been observed between quantum information, quantum chaos and quantum black hole physics [2].

One elegant conjecture that has arisen out of the study of black holes [3] is the fast scrambling conjecture [2]: the time $t_s$ it takes to "scramble" quantum information in a system with $N \gg 1$ degrees of freedom and few-body interactions scales as

$$t_s \gtrsim \frac{\log N}{\gamma}. \tag{1}$$

This conjecture is intended to hold in all many-body quantum systems with few-body interactions. At finite temperature $T$, it is believed that $\gamma \lesssim k_B T / \hbar$. This conjecture ought to hold in quantum systems whose Hilbert space is a tensor product

$$\mathcal{H} = \bigotimes_{i=1}^{N} \mathcal{H}_i, \tag{2}$$

and whose Hamiltonian can be written as a sum of Hermitian operators, each of which acts non-trivially on a finite subset of the degrees of freedom $1, \ldots, N$.

In our view, there is no unambiguous definition of what it means to "scramble" quantum information. A fairly strong definition of scrambling is based on how fast an initially unentangled many-body state can become highly entangled [2]. Suppose that the system is prepared in an initially unentangled state $|\Psi_1\rangle \otimes |\Psi_2\rangle \otimes \cdots \otimes |\Psi_N\rangle$ at time $t = 0$. For any subset $A \subset \{1, \ldots, N\}$ consisting of about half of the degrees of freedom, the entanglement entropy of those degrees of freedom with the remaining ones vanishes: $S_A(t = 0) = 0$. The scrambling time is the minimal time at which $S_A(t_s) \approx \log \dim \mathcal{H}_A - a$, where $\dim \mathcal{H}_A$ is the dimension of the Hilbert space of $A$, and $a > 0$ is an O(1) constant offset. With this definition, there are no known many-body systems with few-body interactions which violate (1) [4].

A more popular definition of scrambling is via out-of-time-ordered correlators (OTOCs) [5]: if $\mathcal{O}_i$ denotes a local operator on vertex $i$, then we expect that

$$\left| \frac{\langle [\mathcal{O}_i(t), \mathcal{O}_j]^2 \rangle}{\langle \mathcal{O}_i^2 \rangle \langle \mathcal{O}_j^2 \rangle} \right| \sim 1, \quad \text{only when } t \gtrsim t_s. \tag{3}$$

The motivation for (1) becomes that in a many-body chaotic system, one might expect [5]

$$\langle [\mathcal{O}_i(t), \mathcal{O}_j]^2 \rangle \lesssim \frac{1}{N} e^{\lambda_L t}, \tag{4}$$

through an analogy to classical chaos. However, this is a weaker notion of scrambling. There exist quantum dynamical systems (random unitary circuits on certain graphs) in which OTOCs grow faster than exponentially, and $t_s \propto N^0$ [4]. Whether or not $t_s \propto N^0$ is possible in quantum systems with a time-independent Hamiltonian has remained an open problem. Of particular interest is the fate of the

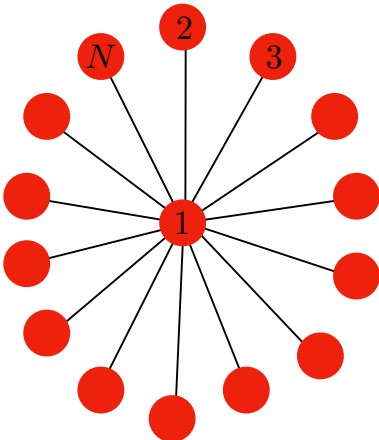

**Figure 1:** A depiction of the star graph. The Hamiltonian (7) only couples spins connected by an edge in the graph.

"chaos bound" which suggests that if $\langle\cdots\rangle$ in (4) represents a suitably regularized thermal correlation function, [6]

$$\lambda_{\mathrm{L}} \leq 2\pi T. \tag{5}$$

Here and henceforth, we set $\hbar = k_{\mathrm{B}} = 1$.

The purpose of this paper is to present a family of Hamiltonian quantum many-body systems with highly unusual quantum information dynamics. This system serves as a stress test for the fast scrambling conjecture, and we will explicitly construct a model where (1) and (3) do not hold for any initially small operator. We consider quantum systems where $\dim(\mathcal{H}_i) = 2$. Let $X_i$, $Y_i$ and $Z_i$ denote the three Pauli matrices acting on $\mathcal{H}_i$, collectively denoted with $X_i^\alpha$ ($\alpha = 1, 2, 3$):

$$X = \begin{pmatrix} 0 & 1 \\ 1 & 0 \end{pmatrix}, \quad Y = \begin{pmatrix} 0 & -\mathrm{i} \\ \mathrm{i} & 0 \end{pmatrix}, \quad Z = \begin{pmatrix} 1 & 0 \\ 0 & -1 \end{pmatrix}. \tag{6}$$

We study quantum mechanical systems whose Hamiltonian is

$$H = \sum_{i=2}^N J_i^{\alpha\beta} X_i^\alpha X_1^\beta + \sum_{i=1}^N B_i^\alpha X_i^\alpha. \tag{7}$$

These Hamiltonians are 2-local, in the computer science sense. All spins interact on a *star graph* – namely, the central spin 1 is connected to all others $2, \ldots, N$, while all outer spins $2, \ldots, N$ only connect to 1: see Figure 1.

The unusual quantum dynamics in this model provide an explicit counterexample to the fast scrambling conjecture in the form (3). Operators can grow large extremely quickly in this system; as the bound on chaos (5) can be understood as a bound on operator growth, we immediately find "violations" of the inequality (5). We demonstrate this exactly in an integrable model, and provide analytic arguments and numerical support for this result in general chaotic models. In the chaotic model, OTOCs grow rapidly for arbitrary choices of initially small operators $\mathcal{O}_{i,j}$. So long as the chaotic phase persists to any finite temperature, OTOCs in our model are unbounded: (4) and (5) will remain invalid. Based on numerical evidence in this paper, we propose that our model is chaotic at sufficiently high (but finite) temperature, so that (5) does not apply. In fact, the model (7) does not obey a key assumption necessary [6] in the proof of (5), so there is no contradiction with their theorem as formally stated. However, to the

extent that black holes and related systems are meant to be the fastest "scramblers", a system which can parametrically violate (5) is quite surprising.

Rapid operator growth on the star graph is a prediction of simple infection-like cartoons of quantum chaos and operator growth [4]. However, we will see that operator growth dynamics on the star graph is uniquely *quantum*. It is not properly modeled by stochastic (and effectively classical) models of (quantum) many-body chaos and operator growth, such as Brownian Hamiltonian evolution [7, 8], or random unitary circuits [9, 10]. In these stochastic models, operators grow rapidly on the star graph because they explosively grow outwards upon reaching the center ($j = 1$ vertex). Yet in quantum mechanics, such rapid operator growth from the center actually stops quantum operators from growing at the edges ($j = 2, \ldots, N$ vertices). This is analogous to the quantum Zeno effect [11]: because probability amplitudes add in quantum mechanics (and not probabilities), quantum information is protected by coupling it to a rapidly varying source (or a measuring device, as in the canonical example). Ultimately, our simple models on the star graph will serve as a useful testing round for understanding fundamental constraints and limitations on quantum information dynamics in many-body quantum systems with few-body ($k$-local) interactions.

## 2 Thermodynamics

In this section, we take a small detour from our main focus on operator growth and quantum information dynamics. The primary purpose of this section is to justify that the Hamiltonian (7) is generically not integrable at infinite temperature. Nevertheless, we will also take the opportunity to make some simple and preliminary comparisons with random matrix theory, and to explore the finite temperature physics of the model.

### 2.1 Extensivity

Our first goal is to confirm that the thermodynamics and spectrum of the model will be sensible (namely, the free energy $F \propto N$) if all of the $J_j^{\alpha\beta}$ and $B_i^\alpha$ scale as $N^0$. We begin by studying a simplified, integrable version of (7):

$$H = \sum_{i=2}^{N} J_i Z_i Z_1. \tag{8}$$

This is the Ising Hamiltonian on the star graph, and will be referred to as such below. The partition function is classical:

$$Z(\beta) = \text{tr}\left[e^{-\beta H}\right] = \sum_{Z_j=\pm 1} \prod_{i=2}^{N} e^{-\beta J_i Z_i Z_1} = \sum_{Z_1=\pm 1} \prod_{i=2}^{N} \left(\sum_{Z_i=\pm 1} e^{-\beta J_i Z_i Z_1}\right) = 2\prod_{i=2}^{N} \left(2\cosh(\beta J_i)\right). \tag{9}$$

with $\beta = 1/T$ the inverse temperature. The free energy is

$$F = -T \log Z \approx -NT\mathbb{E}\left[\log\left(2\cosh\frac{J}{T}\right)\right] \tag{10}$$

where $\mathbb{E}[\cdots]$ denotes the average over couplings $J_i$. Note that $J_i$ may be taken to be random variables, or to be fixed. As we will see, the model is a bit nicer if $J_i$ are random variables. But regardless of our choice, we see that $J_i \propto N^0$ is the correct scaling so that $H$ is extensive in the thermodynamic limit.

For more generic couplings, we can confirm numerically that the spectrum is extensive if $J_i^{\alpha\beta} \propto N^0$. But there is also a simple variational argument. Let $|\Psi\rangle = |+\rangle_1 \otimes |\psi_2\rangle \otimes \cdots \otimes |\psi_N\rangle$ be a many-body wave

function, with $Z_1|+\rangle_1 = |+\rangle_1$ and $|\psi_j\rangle \in \mathcal{H}_j$. Now consider

$$\langle \Psi | H | \Psi \rangle = B_1^Z + \sum_{j=2}^{N} \sum_{\alpha=1}^{3} \left( J_j^{Z\alpha} + B_j^{\alpha} \right) \langle \psi_j | X_j^{\alpha} | \psi_j \rangle \tag{11}$$

By choosing the states $|\psi_j\rangle$ to be eigenvectors of $J_j^{Z\alpha} X_j^{\alpha}$ we find $2^{N-1}$ orthogonal states whose average energies are

$$\langle E(\sigma_2, \ldots, \sigma_N) \rangle = B_1^Z + \sum_{j=2}^{N} \sigma_j \sqrt{\sum_{\alpha=1}^{3} \left( J_j^{Z\alpha} \right)^2} \tag{12}$$

Here $\sigma_j \in \{\pm 1\}$ for $2 \leq j \leq N$. If $J_j^{\alpha\beta}$ are O(1), (12) guarantees the existence of eigenstates of $H$ with eigenvalue $\propto N$. Using the triangle inequality for operators, $\|A + B\| \leq \|A\| + \|B\|$, it is easy to see that $H$ cannot have any eigenvalues that scale faster than $N$. Since (12) implies that the spectrum is extensive with $J_j^{\alpha\beta} \propto N^0$, this is the scaling we will take henceforth.

## 2.2 | Level Statistics

Contrary to what (12) implies, this model is *not generically integrable*. A simple test for integrability is to study the spacing of nearby eigenstates of the Hamiltonian $H$. We study a random ensemble of Hamiltonians of the form (7), with $J_i^{\alpha\beta}$ and $B_i^{\alpha}$ all taken to be independent Gaussian zero-mean random variables with variance $\mathcal{J}^2$:

$$\mathbb{E}\left[ J_i^{\alpha\beta} \right] = \mathbb{E}\left[ B_i^{\alpha} \right] = 0, \tag{13a}$$

$$\mathbb{E}\left[ J_i^{\alpha\beta} J_j^{\gamma\eta} \right] = \mathcal{J}^2 \delta_{ij} \delta^{\alpha\gamma} \delta^{\beta\eta}, \tag{13b}$$

$$\mathbb{E}\left[ B_i^{\alpha} B_j^{\gamma} \right] = \mathcal{J}^2 \delta_{ij} \delta^{\alpha\gamma}. \tag{13c}$$

This Hamiltonian has no residual symmetries and, if it is chaotic, its spectrum will locally be identical to the Gaussian unitary ensemble (GUE) [12].

A simple check for GUE statistics was proposed in [13, 14]. We calculate

$$\bar{r} = \frac{1}{2^N - 2} \sum_{\alpha=2}^{2^N - 1} \frac{\min(E_\alpha - E_{\alpha-1}, E_{\alpha+1} - E_\alpha)}{\max(E_\alpha - E_{\alpha-1}, E_{\alpha+1} - E_\alpha)} \tag{14}$$

numerically on star graphs of relatively small sizes $6 \leq N \leq 12$. Here $E_\alpha$ denote eigenvalues of $H$, ordered as $E_1 < E_2 < \cdots < E_{2^N}$. The prediction of GUE statistics is that $\bar{r} \approx 0.60$. Figure 2 shows that the generic 2-local model on the star graph has $\bar{r}$ compatible with GUE statistics in the thermodynamic limit. In the numerics of this section, GUE random matrices of size $2^N \times 2^N$ were numerically generated in order to compare with our model, as this (by construction) accounts for finite size effects.

However, upon closer inspection, the actual distribution of level statistics is not GUE. Figure 3a plots $P(s)$, the probability distribution of the normalized level spacing

$$s_\alpha = (E_\alpha - E_{\alpha-1}) \frac{2^{N-1} - 1}{E_{2^N} - E_1}. \tag{15}$$

There are clear discrepancies between the GUE prediction and the star graph model. However, upon closer inspection, the differences are somewhat subtle. Figure 3b plots the same probability distributions, but studying $P(s)$ on a logarithmic scale *and* rescaling $s$ by a factor of 1.33 in the GUE prediction. Now

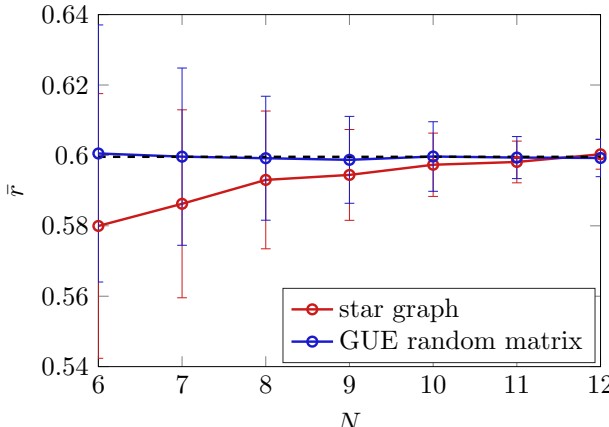

**Figure 2:** The fully random 2-local model on the star graph is not integrable: $\bar{r}$ is comparable between the star graph model (red) and GUE random matrix theory (blue). The numerical result of [14] is the dashed black line.

the small $s$ behavior of the GUE model and the star graph are in close agreement – the curves only differ at large $s$.

We have also simulated level statistics for a model which is widely believed to be a spin glass at zero temperature: the SU(2) Heisenberg model on the complete graph $K_N$ [15]:

$$H = \sum_{\alpha=1}^{3} \sum_{i=1}^{N} B_i^\alpha X_i^\alpha + \frac{1}{\sqrt{N}} \sum_{\alpha=1}^{3} \sum_{i<j} J_{ij} X_i^\alpha X_j^\alpha, \tag{16}$$

with independent Gaussian random couplings:

$$\mathbb{E}\left[J_{ij}\right] = \mathbb{E}\left[B_i^\alpha\right] = 0, \tag{17a}$$

$$\mathbb{E}\left[J_{ij} J_{kl}\right] = \mathcal{J}^2 \delta_{ik} \delta_{kl}, \tag{17b}$$

$$\mathbb{E}\left[B_i^\alpha B_j^\gamma\right] = \mathcal{J}^2 \delta_{ij} \delta^{\alpha\gamma}. \tag{17c}$$

Normally this model is studied without a random field, but we have included the random fields to ensure that there are no residual symmetries. For simplicity we have only looked at one value of the 1-local field strengths. Rather amusingly, the level statistics of this model, without any rescaling, are qualitatively identical to our model on the star graph. At the end of this section, we will return and give a physical cartoon for all of these results.

## 2.3 ▎ Finite Temperature

We now discuss the physics of this model at finite temperature $T = 1/\beta$. The object we will study is the spectral form factor [16]: defining the complex partition function

$$Z(\beta + \mathrm{i}t) \equiv \mathrm{tr}\left(\mathrm{e}^{-(\beta+\mathrm{i}t)H}\right), \tag{18}$$

we will study

$$F(\beta, t) = \left|\frac{Z(\beta + \mathrm{i}t)}{Z(\beta)}\right|^2, \tag{19}$$

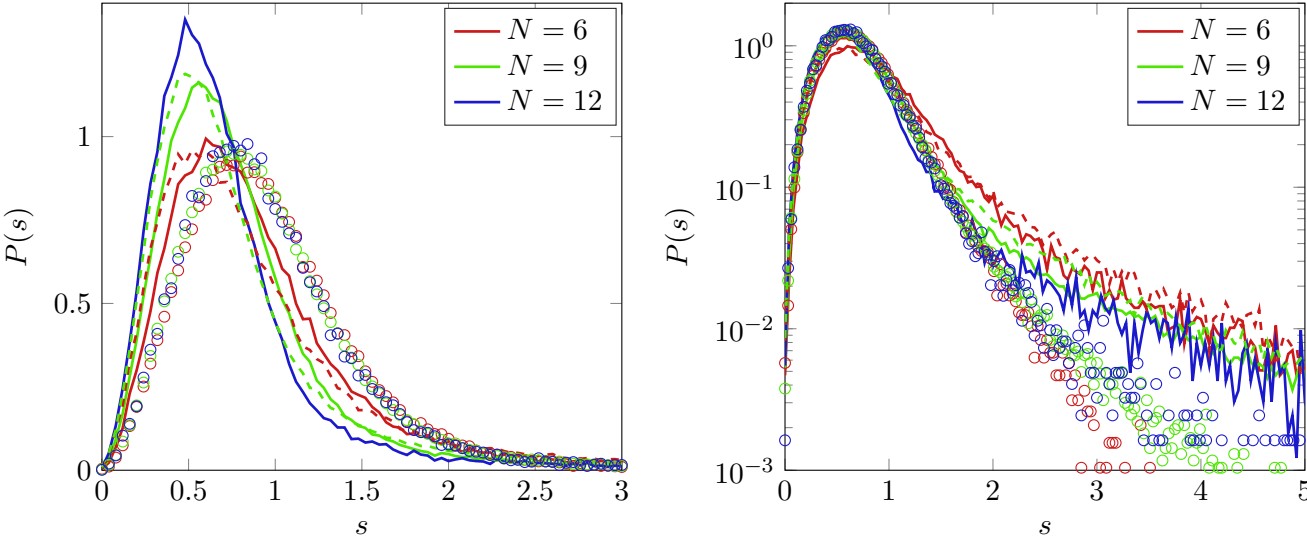

**Figure 3:** Left: the level spacing distribution $P(s)$ in the star graph model (solid line), random-field XYZ model on the complete graph (dashed line), and GUE random matrix (circles). Right: the same plot, now with logarithmic scaling of $P(s)$ *and* a rescaling of $P_{\mathrm{GUE}}(s) \to \alpha P_{\mathrm{GUE}}(\alpha s)$ with $\alpha = 1.33$ for the GUE random matrix. At large $N$ all three models agree for $s \lesssim 1$, while the 2-local random spin models have a heavier tail in the distribution at large $s$. In all data points, $P(s)$ is averaged over a few hundred disorder realizations.

averaged over realizations of disorder. Figure 4 plots the results at $\beta\mathcal{J} = 0$ and 0.4, which appear qualitatively different. At infinite temperature $\beta\mathcal{J} = 0$, much of the structure in $F$ is identical between GUE and our model: at early times, $F$ is dominated by the dephasing of many-body eigenstates. At an intermediate time scale we see that $F$ reaches a minimum, after which it increases again until saturating at a value of order $2^{1-N}$ as $t \to \infty$. The increase is caused by eigenvalue repulsion in a random matrix, as described cleanly in [16]. In fact, the only quantitative disagreements between GUE and the star graph model occur just before the dip time, and we will not explore their origin in this paper.

At finite temperature $\beta\mathcal{J} = 0.4$, it is clear from Figure 4 that the GUE no longer reproduces the star graph model. We now argue that this is a hint of an "integrable phase" at finite temperature. The two key differences between the GUE and the star graph model at finite $T$ are $(i)$ the steepness of the initial decrease in $F(\beta, t)$ is greatly increased, and the early time behavior of $F$ is dependent strongly on $N$, and $(ii)$ the "ramp" at later times is much weaker. These effects are qualitatively observed in another numerical study of a disorder averaged spectral form factor in an integrable model [17]. We also note that while at infinite temperature $F$ is also $N$-dependent at early times, this is true for both the random matrix and the star graph model, and may be a consequence of very strong finite size effects at high temperatures. Henceforth, we will loosely refer to this integrable phase as a "quantum spin glass" but we are not sure that this phase is, in fact, a glass (though quantum fluctuations are likely important), and/or whether it is many body localized [18].

It was recently argued that many features of the spectral form factor can be mimicked in an integrable theory upon averaging over random couplings [17]. In Figure 4 this is manifest in the continued presence of the dip/ramp/plateau structure in $F(t)$ in the glass phase. In order to more carefully distinguish between a chaotic phase and a non-chaotic "glass" phase, it is useful to plot $F(\beta, t)$ for a *single realization*

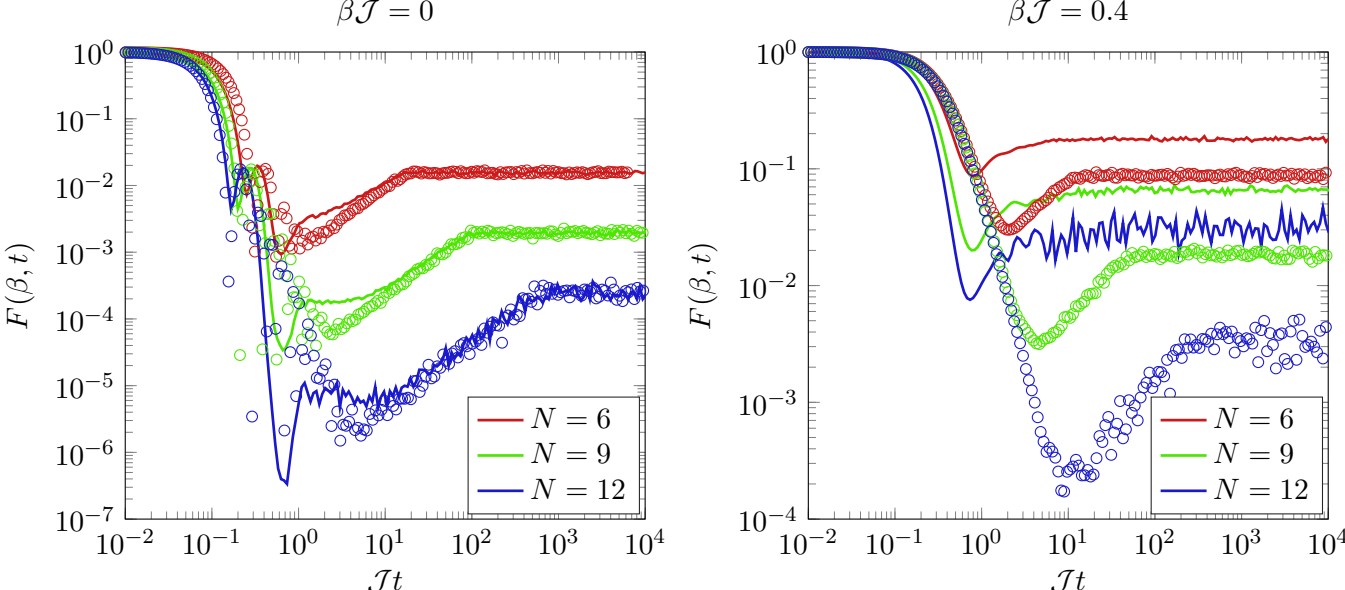

**Figure 4:** $F(\beta, t)$ at infinite and finite temperature in the star graph model (solid lines) and GUE random matrix theory (circles). We have normalized the random matrices so that the early time behavior of $F(\beta, t)$ matches betweeen the two theories at infinite temperature. A few hundred realizations of disorder are used to disorder average. While both theories appear similarly "chaotic" at $\beta\mathcal{J} = 0$, by $\beta\mathcal{J} = 0.4$ it is clear that the two models are distinguishable.

of disorder. We do so in Figure 5. As a trivial example of an integrable phase with which we may contrast the chaotic random matrix model, we consider the 1-local model

$$H = \sum_{j=1}^{N} B_j^Z Z_j, \tag{20}$$

with $B_j^Z$ independent Gaussian random variables, normalized as in (17). The key observation is that the ramp in $F(t)$ at intermediate time scales is absent in integrable phases; indeed, eigenvalues are essentially uncorrelated in an integrable phase [13]. Even on a single realization the ramp is clearly visible in both GUE and the star graph model, while not in the 1-local model, at $\beta\mathcal{J} = 0$. By $\beta\mathcal{J} = 0.4$, the ramp structure is mostly absent from the star graph model, except possibly in a very short time window (one decade or less); in contrast, the GUE realization still has a clear ramp over two decades of time.

Our proposal that there is a thermal phase transition in our model (to a glassy phase) also resolves our earlier puzzle with the normalization of the level statistics distribution $P(s)$. We propose that in the large $N$ limit, the eigenvalue spectrum of the model on the star graph schematically consistents of a large bulk, which is locally random-matrix-like, flanked on the tails (both at the highest and lowest energies) by widely spaced eigenstates which are not random-matrix-like, even locally. A fraction $\mathrm{e}^{-\eta N}$ ($\eta > 0$) of eigenstates lie in the tails, in the large $N$ limit. Hence $P(s)$ becomes

$$P(s) = \alpha \left(1 - \mathrm{e}^{-\eta N}\right) P_{\mathrm{GUE}}(\alpha s) + \mathrm{e}^{-\eta N} P_{\mathrm{IR}}(s), \tag{21}$$

where $\eta > 0$ is a finite, $\mathrm{O}(N^0)$ coefficient and $\alpha \approx 1.33$ as determined numerically. $P_{\mathrm{GUE}}$ and $P_{\mathrm{IR}}$ are normalized probability distributions, with $P_{\mathrm{IR}}$ having a "heavy tail". We predict $P_{\mathrm{IR}}(s)$ to be a Poisson

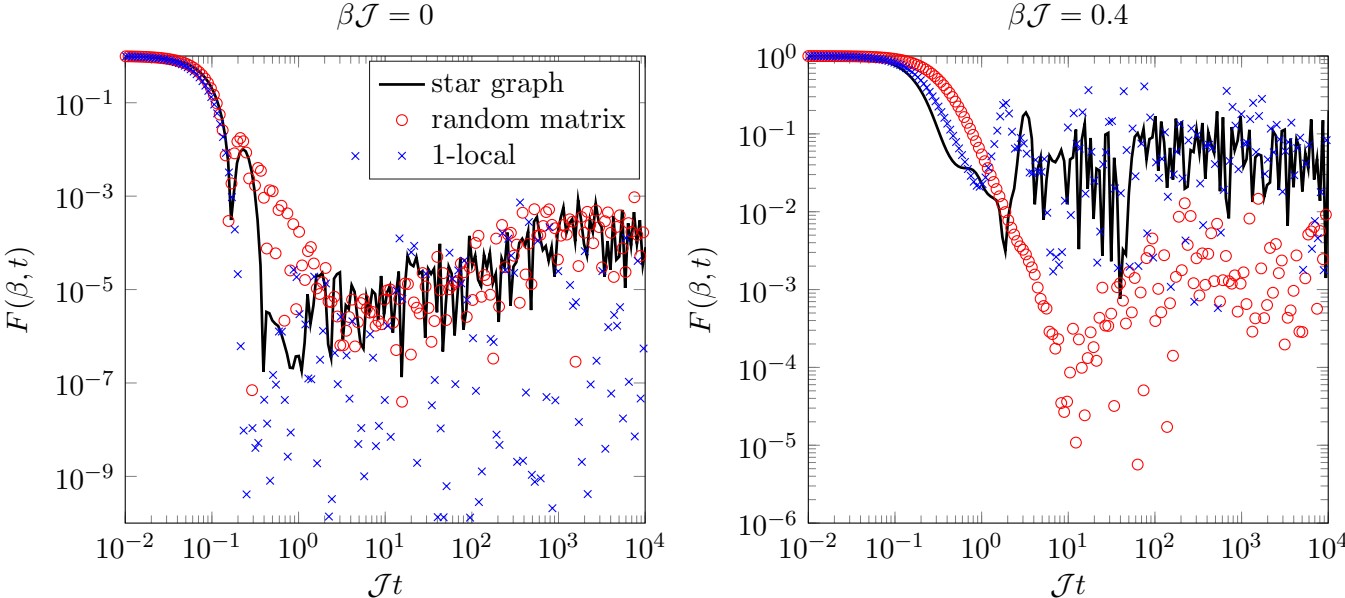

**Figure 5:** $F(\beta, t)$ at infinite and finite temperature in the star graph model (solid lines), GUE random matrix theory (circles), and an integrable 1-local model (crosses) whose field strengths are drawn from an $N \times N$ random matrix. Only one disorder realization is used for each. In integrable phases, we see that the ramp structure is lost and is replaced with extremely oscillatory behavior, which is weakened at finite temperature. The star graph model is qualitatively similar to GUE at $\beta\mathcal{J} = 0$, and to an integrable model at $\beta\mathcal{J} = 0.4$.

distribution: $P_{\mathrm{IR}}(s) \propto \exp(-\lambda s)$ for $s > 1$, which would be associated with an integrable phase. It is $P_{\mathrm{IR}}$ which controls the glassy physics at low temperature, and is responsible for $\alpha > 1$.

As an extreme example illustrating the above point, suppose that $\eta = 2$, that $2^N - 1$ eigenstates directly come from a GUE random matrix with eigenvalues in the band $[-N\mathcal{J}, N\mathcal{J}]$, and that the last eigenstate is at energy $-N\mathcal{J}_0$, with $\mathcal{J}_0 > \mathcal{J}$. The partition function is (to leading order in the exponentials)

$$Z(\beta) = 2^N \mathrm{e}^{N\beta\mathcal{J}} + \mathrm{e}^{N\beta\mathcal{J}_0}; \tag{22}$$

the free energy is

$$F(\beta) = N \min(-\mathcal{J}_0, -\mathcal{J} - T\log 2). \tag{23}$$

Below the critical temperature $T_{\mathrm{c}} = \frac{1}{\log 2}(\mathcal{J}_0 - \mathcal{J})$, the system collapses into the ground state and there is no chaotic dynamics left. Furthermore, the presence of this low lying state modifies the mean level spacing: as seen from (15), the parameter $\alpha$ in (21) is given by

$$\alpha = \frac{\mathcal{J} + \mathcal{J}_0}{2\mathcal{J}}. \tag{24}$$

In reality, the model on the star graph is not controlled by a single low lying eigenstate. Numerically, we find that $\max(E_\alpha - E_{\alpha-1}) \approx 2\mathcal{J}$ in (7), with random couplings (13): this result approximately holds for any $N$ and a generic instance of couplings. Such a level spacing among a polynomial number of states is adequate to drive a phase transition at finite temperature.

Our numerics on $F$ are not powerful enough to resolve the transition temperature to a postulated glassy phase or even probe its $N$-dependence. Beyond the comments made above, we leave the precise

realization of the phase diagram of our model to another paper. The point of the discussion here is simply to emphasize that the model is effectively chaotic at infinite temperature, but likely not at sufficiently low temperature. Whether or not the unusual operator dynamics we describe below necessarily implies a glassy phase at low temperature is an interesting question which we leave open.

## 3 Operator Dynamics with Integrability

Having established that our models on the star graph are generically chaotic, we will now postpone chaos for one more section. As it turns out, many critical elements of operator growth on the star graph can be seen in simpler integrable models with 2-local couplings. As analytic results are available in this limit, we start the discussion here.

### 3.1 Operator Growth in the Ising Model

First, we will exactly describe the time evolution of all operators for all times in the Ising model [19, 20]. The key observation is that all terms in the Hamiltonian mutually commute. Let $\mathcal{L}\mathcal{O} = \mathrm{i}[H, \mathcal{O}]$, and

$$\mathcal{L}_i \mathcal{O} = \mathrm{i}[J_i Z_i Z_1, \mathcal{O}] \tag{25}$$

for any operator $\mathcal{O}$; hence $\mathcal{L} = \sum_{j=2}^{N} \mathcal{L}_j$. Since

$$[\mathcal{L}_i, \mathcal{L}_j] = 0, \tag{26}$$

operators evolve with time as

$$\mathcal{O}(t) = \prod_{j=2}^{N} \mathrm{e}^{\mathcal{L}_j t} \mathcal{O}. \tag{27}$$

The product can be ordered in any way.

It is often useful to think about Hermitian operators $\mathcal{O}$ as spanning a real vector space with inner product

$$(\mathcal{O}_1 | \mathcal{O}_2) = 2^{-N} \mathrm{tr}\left(\mathcal{O}_1 \mathcal{O}_2\right). \tag{28}$$

An orthonormal basis for this vector space is $\bigotimes_{i=1}^{N} \{1_i, X_i, Y_i, Z_i\}$. On this vector space, $\mathcal{L}$ is an antisymmetric matrix, as time evolution in quantum mechanics is unitary. Hence,

$$\mathrm{tr}(\mathcal{O}^2) = 4^{-N} \sum_{\text{basis } \mathcal{O}_\alpha} \mathrm{tr}(\mathcal{O}_\alpha \mathcal{O}(t))^2 \tag{29}$$

for any Hermitian operator $\mathcal{O}$. The statements of this paragraph are true for any Hermitian $H$.

Our goal is now to explicitly evaluate the time evolution of all Hermitian operators in the model (8). A useful trick is to look at the time evolution of non-Hermitian operators

$$X_i^{\pm} = X_i \pm \mathrm{i}Y_i. \tag{30}$$

Suitable linear combinations of these operators generate $X_i$ and $Y_i$. First, let us consider the case $i = 2$. Since $\mathcal{L}_j X_2^{\pm} = 0$ for $j \neq 2$, using (27) we conclude that

$$X_2^{\pm}(t) = \mathrm{e}^{\mathcal{L}_2 t} X_2^{\pm}. \tag{31}$$

Next, we observe that

$$[\mathrm{i}Z_1 Z_2, X_2^{\pm}] = \mathrm{i}Z_1 \otimes (2\mathrm{i}Y_2 \mp 2\mathrm{i}X_2) = \pm 2\mathrm{i}Z_1 X_2^{\pm} \tag{32}$$

Thus

$$X_2^\pm(t) = e^{\pm 2iJ_2 Z_1 t} X_2^\pm = (\cos(2J_2 t) \pm i \sin(2J_2 t) Z_1) X_2^\pm. \tag{33}$$

More generally, if $\sigma_j \in \{\pm 1\}$, then for any subset $A \subset \{2, \ldots, N\}$,

$$e^{\mathcal{L}t} \prod_{i \in A} X_i^{\sigma_i} = \prod_{i \in A} X_i^{\sigma_i}(t), \tag{34}$$

and there are no non-commuting operators in the product above.

Next, let us consider the operator $X_1^\pm(t)$. In this case we can again use (32) to find

$$\prod_{j=2}^N e^{\mathcal{L}_j t} X_1^\pm = \prod_{j=3}^N e^{\mathcal{L}_j t} \left(e^{\pm 2iJ_2 Z_1 t} X_1^\pm\right) = X_1^\pm \prod_{j=2}^N e^{\pm 2iJ_j Z_1 t} \tag{35}$$

Hence $X_j^\pm$ grows to be a many-body operator (albeit in a trivial way) after a finite time $t \sim J^{-1}$.

Lastly, we consider operators of the form $X_1^\pm \prod_{i \in A} X_i^{\sigma_i}$; again $A \subset \{2, \ldots, N\}$. In this case, we use the fact that

$$[Z_1 Z_j, X_1^\pm X_j] = [Z_1 Z_j, X_1^\pm Y_j] = 0 \tag{36}$$

to find that

$$\left(X_1^\pm \prod_{i \in A} X_i^{\sigma_i}\right)(t) = X_1^\pm \prod_{i \in A} X_i^{\sigma_i} \prod_{i \notin A} e^{\pm 2iJ_j Z_1 t}. \tag{37}$$

Since $H$ commutes with arbitrary products of $Z_i$, the time evolution of any operator with any $Z$s is trivially found by multiplying one of the answers above by appropriate $Z$. So using (33), (34) and (37), we have found the exact time evolution of all operators.

## 3.2 ▏ Finite Temperature in the Ising Model

In (35), we found that operators can grow large quickly: after O(1) time, most weight in $X_1(t)$ will be in terms which contain O($N$) Pauli matrices. At any temperature, this observation has important consequences: both "regulated" and "unregulated" thermal out-of-time-ordered correlators (OTOCs) grow large in O(1) time. Let $\rho = \frac{1}{Z} e^{-\beta H}$ denote the thermal density matrix; then the unregulated OTOC

$$\mathrm{tr}\left[\rho[X_1(t), X_2]^2\right] = \mathrm{tr}\left[\rho[X_1, X_2(-t)]^2\right] = \mathrm{tr}\left[\rho\left[X_1, \cos(2J_2 t) X_2 - \sin(2J_2 t) Z_1 Y_2\right]^2\right]$$
$$= -4\sin^2(2J_2 t) \mathrm{tr}\left[\rho(Y_1 Y_2)^2\right] = -4\sin^2(2J_2 t). \tag{38}$$

There is no temperature dependence in this result at all. We now turn to the regulated OTOC:

$$C_{12}(t) = \frac{\mathrm{tr}\left[\sqrt{\rho}[X_1(t), X_2]\sqrt{\rho}[X_1(t), X_2]\right]}{\mathrm{tr}[\sqrt{\rho}X_1\sqrt{\rho}X_1]\mathrm{tr}[\sqrt{\rho}X_2\sqrt{\rho}X_2]}. \tag{39}$$

First we evaluate the terms in the denominator, writing out the trace in the basis of mutual eigenvectors of $Z_i$: $|\mathbf{z}\rangle$ for $z_i \in \{\pm 1\}$:

$$\mathrm{tr}(\sqrt{\rho}X_1\sqrt{\rho}X_1) = \sum_{\mathbf{z}} \langle z_1 \cdots z_N|\sqrt{\rho}|z_1 \cdots z_N\rangle\langle(-z_1)\cdots z_N|\sqrt{\rho}|(-z_1)\cdots z_N\rangle$$

$$= \frac{1}{Z(\beta)} \sum_{\mathbf{z}} \exp\left[-\frac{\beta}{2}\sum_{j=2}^N J_j z_1 z_j - \frac{\beta}{2}\sum_{j=2}^N J_j(-z_1)z_j\right] = \frac{2^N}{Z(\beta)}, \tag{40a}$$

$$\mathrm{tr}(\sqrt{\rho}X_2\sqrt{\rho}X_2) = \sum_{\mathbf{z}} \langle z_1 z_2 \cdots z_N | \sqrt{\rho} | z_1 z_2 \cdots z_N \rangle \langle z_1(-z_2) \cdots z_N | \sqrt{\rho} | z_1(-z_2) \cdots z_N \rangle$$

$$= \frac{1}{Z(\beta)} \sum_{\mathbf{z}} \exp\left[-\beta \sum_{j=3}^{N} J_j z_1 z_j\right] = \frac{4}{Z(\beta)} \prod_{j=3}^{N} \left(2\cosh(\beta J_j)\right) = \frac{1}{\cosh(\beta J_2)} \qquad (40\mathrm{b})$$

In the last step, we have used (9). Now evaluating the numerator of (79), using similar tricks as in (38):

$$\mathrm{tr}\left[\sqrt{\rho}[X_1(t), X_2]\sqrt{\rho}[X_1(t), X_2]\right] = \frac{1}{Z(\beta)}\mathrm{tr}\left[\left(\mathrm{e}^{-\beta H/2}2\mathrm{i}\sin(2J_2 t)Y_1 Y_2\right)^2\right]$$

$$= -\frac{4\sin^2(2J_2 t)}{Z(\beta)}\sum_{\mathbf{z}}(\mathrm{i}^2(-z_1)(-z_2))\mathrm{e}^{-\beta H(-z_1,-z_2,\ldots,z_N)/2}(\mathrm{i}^2 z_1 z_2)\mathrm{e}^{-\beta H(z_1,z_2,\ldots,z_N)/2}$$

$$= -\frac{4\sin^2(2J_2 t)}{Z(\beta)}\sum_{\mathbf{z}}\exp\left[-\beta J_2 z_1 z_2 - \frac{\beta}{2}\sum_{j=3}^{N}J_j(z_1 - z_1)z_j\right] = -\frac{4\sin^2(2J_2 t)}{Z(\beta)} \times 2^N \cosh(\beta J_2). \qquad (41)$$

We conclude that
$$C_{12}(t) = -4\sin^2(2J_2 t)\cosh^2(\beta J_2). \qquad (42)$$

Amusingly, the regulated OTOC actually grows a little bit larger than the unregulated OTOC: this is largely due to the fact that the denominator in $C_{12}(t)$ is extremely small.

We conclude that at any temperature, the chaos bound (5) is "violated", in so far as the time scale at which $C_{1j}(t) \sim 1$, simultaneously for all $j$, is not bounded by $\beta \log N$. The origin of this violation is simple. One key assumption in the chaos bound is that $[X_1(t), X_2]$ is small for all times $t \lesssim \beta$ for distinct degrees of freedom [6], yet on the star graph this is not true. While it is known that other integrable models (notably free theories) violate the chaos bound [6], it is possible that the violation above extends to general non-integrable models on the star graph, as we discuss in Section 4. Moreover, an important difference between our model and a free theory is that there exists an O(1) time $t$ at which, for O($N$) values of $j > 1$, $\mathrm{tr}[\rho[X_1(t), X_j]^2]$ is large. This means that $X_1(t)$ genuinely evolves to become a "complicated" many-body operator after a finite amount of time.

Of course, at the same time $X_1(t)$ becomes complicated on a constant time scale, most quantum information in the system is protected. More precisely, operators $X_j$, $Y_j$ and $Z_j$ never decay into complicated operators for all times $t$. At times $t = \pi/2J_j$, $X_j^{\alpha} X_j^{\alpha}(t) = \pm 1$: any information stored in a perturbation of system can be exactly recovered. In the language of OTOCs, $\mathrm{tr}[\rho[X_2(t), X_3]^2] = \mathrm{tr}[(\sqrt{\rho}[X_2(t), X_3])^2] = 0$. While the chaos bound is violated for one choice of operators $X_1$ and $X_2$, it does hold (trivially) for most pairs of single Pauli matrices. This is compatible with the conjecture of [21] that in a generic quantum many-body system, most operators which have a small size (at finite temperature) at $t = 0$ will remain small under Heisenberg time evolution up to time scales $t \lesssim \beta$.

### 3.3 | Adding a Transverse Field

Next, we modify (8) to a random transverse field Ising model:

$$H = BX_1 + \sum_{i=2}^{N} J_i Z_i Z_1, \qquad (43)$$

where $B$ is a perturbatively small parameters. This model is still integrable: we still have $[H, Z_j] = 0$ for $j \neq 1$, and can exactly diagonalize $H$. The energy levels are given by

$$E(Z_2, \ldots, Z_N) = \pm \sqrt{B^2 + \left( \sum_{j=2}^{N} J_j Z_j \right)^2}. \tag{44}$$

Nevertheless, we present a calculation of $\mathrm{tr}(X_2(t)X_2)$, and observe that it decays at late times, whenever the couplings $J_j$ are sufficiently spread. It is useful to invoke the memory matrix formalism [22], which allows us to take the Heisenberg picture evolution equation $\partial_t|\mathcal{O}) = \mathcal{L}|\mathcal{O})$ and integrate out all operators in the vector space of operators except $|X_2)$ and $|Z_1Y_2)$. We will see that this calculation provides an accurate and simple characterization of the early time behavior of $\mathrm{tr}(X_2(t)X_2)$, while subtleties arise at late times in this integrable model. We let

$$a(t) = (X_2|X_2(t)), \tag{45a}$$
$$b(t) = (Z_1Y_2|X_2(t)) \tag{45b}$$

denote the coefficients of the evolving operator $X_2(t)$ in the $X_2$ and $Y_2Z_1$ directions in operator Hilbert space. Defining the projectors

$$\mathfrak{p} = |X_2)(X_2| + |Z_1Y_2)(Z_1Y_2| = 1 - \mathfrak{q}, \tag{46}$$

and using the exact identity

$$\frac{\mathrm{d}}{\mathrm{d}t}\mathfrak{p}|\mathcal{O}(t)) = \mathfrak{p}\mathcal{L}\mathfrak{p}|\mathcal{O}(t)) + \int_0^t \mathrm{d}s \, \mathfrak{p}\mathcal{L}\mathfrak{q}e^{\mathfrak{q}\mathcal{L}\mathfrak{q}s}\mathfrak{q}\mathcal{L}\mathfrak{p}|\mathcal{O}(s)) \tag{47}$$

which holds as $\mathfrak{p}|X_2(0)) = |X_2)$:

$$\frac{\mathrm{d}}{\mathrm{d}t} \begin{pmatrix} a(t) \\ b(t) \end{pmatrix} = \begin{pmatrix} 0 & -2J_2 \\ 2J_2 & 0 \end{pmatrix} \begin{pmatrix} a(t) \\ b(t) \end{pmatrix} - \int_0^t \mathrm{d}s \begin{pmatrix} 0 & 0 \\ 0 & \mathcal{K}(t-s) \end{pmatrix} \begin{pmatrix} a(s) \\ b(s) \end{pmatrix}, \tag{48}$$

Here, the kernel $\mathcal{K}(t-s)$ arises due to the fluctuations of the modes which have been integrated out: to leading order in $B$,

$$\mathcal{K}(s) = \mathrm{tr}\left(Y_2Z_1\left(-\mathcal{L}_B e^{i\mathcal{L}_0 s}\mathcal{L}_B Y_2 Z_1\right)\right) = 4B^2 \mathrm{tr}\left(Y_2Y_1(Y_2Y_1)(s)\right) = 4B^2 \prod_{j=3}^{N} \cos(2J_j s) \tag{49}$$

where $\mathcal{L}_0$ denotes commutation with $H$, evaluated at $B = 0$, and $\mathcal{L}_B\mathcal{O} = iB[X_0, \mathcal{O}]$.

For simplicity, let us now assume that $J_i$ are independent, identically distributed Gaussian random variables of mean zero and variance $\mathcal{J}^2$. In this case, we may safely disorder average

$$\mathbb{E}[\mathcal{K}(s)] = 4B^2\mathbb{E}[\cos(2J_j s)]^{N-2} \approx 4B^2 e^{-2N\mathcal{J}^2 s^2}. \tag{50}$$

To show that fluctuations between different realizations of disorder are negligible:

$$\frac{\mathbb{E}[\mathcal{K}(s)^2]}{\mathbb{E}[\mathcal{K}(s)]^2} \approx \cosh^N(4\mathcal{J}^2 s^2); \tag{51}$$

for times

$$t \ll t_* = \frac{1}{N^{1/4}\mathcal{J}} \tag{52}$$

statistical fluctuations in $\mathcal{K}(t)$ are negligible. Since $\mathcal{K}(t_*) \propto \mathrm{e}^{-\sqrt{N}}$ we will replace $\mathcal{K}(s)$ with $\mathbb{E}[\mathcal{K}(s)]$ henceforth and drop the explicit and negligible disorder average.

The kernel $\mathcal{K}(s)$ decays extremely fast, so it is accurate to approximate

$$\int\limits_0^t \mathrm{d}s \, \mathcal{K}(t-s)b(s) \approx b(t) \int\limits_0^\infty \mathrm{d}s \, \mathcal{K}(s) = \sqrt{\frac{2\pi}{N}} \frac{B^2}{\mathcal{J}} b(t). \tag{53}$$

Combining (48) and (53) we obtain

$$2^{-N}\mathrm{tr}(X_2(t)X_2) \approx \cos(2J_2 t)\exp\left[-\sqrt{\frac{\pi}{2N}}\frac{B^2}{\mathcal{J}}t\right] \tag{54}$$

So we predict that $X_2(t)$ decays over the time scale

$$t_{\mathrm{coh}} = \sqrt{\frac{2N}{\pi}}\frac{\mathcal{J}}{B^2} \tag{55}$$

in the thermodynamic limit.

Crucially, the origin of the long coherence time (55) is the rapid growth of operators (37) on the central vertex. A simple way of understanding this effect is by recognizing that just like in ordinary quantum mechanics, the "probabilities" of finding an operator in a given "state" add quadratically (29). From first order perturbation theory,

$$(Z_1 Y_2)(t) = \cos(2J_2 t)Z_1 Y_2 - \sin(2J_2 t)X_2 + 2B\int\limits_0^t \mathrm{d}s \, \cos(2J_2(t-s))(Y_1 Y_2)(s) + \mathrm{O}(B^2). \tag{56}$$

We now estimate the weight of the growing operator $(Z_1 Y_2)(t)$ contained in the first order term:

$$4B^2 \int\limits_0^t\int\limits_0^t \mathrm{d}s_1 \mathrm{d}s_2 \cos(2J_2(t-s_1))\cos(2J_2(t-s_2))((Y_1 Y_2)(s_1)|(Y_1 Y_2)(s_2))$$

$$= \int\limits_0^t\int\limits_0^t \mathrm{d}s_1 \mathrm{d}s_2 \cos(2J_2(t-s_1))\cos(2J_2(t-s_2))\mathcal{K}(s_1-s_2)$$

$$\approx 4B^2 \int\limits_0^t\int\limits_0^t \mathrm{d}s_1 \mathrm{d}s_2 \cos(2J_2(t-s_1))\cos(2J_2(t-s_2))\mathrm{e}^{-2N\mathcal{J}^2(s_1-s_2)^2} \approx 2\sqrt{\frac{2\pi}{N}}\frac{B^2}{\mathcal{J}}\int\limits_0^t \mathrm{d}s \cos^2(2J_2(t-s))$$

$$\approx \sqrt{\frac{2\pi}{N}}\frac{B^2}{\mathcal{J}}t. \tag{57}$$

The approximations above are sensible at sufficiently early times $t$ where the coefficient above is much smaller than 1. We conclude that, somewhat counterintuitively, the growing operator $(Y_1 Y_2)(s)$ grows so quickly that it actually *prevents* the operator $(Z_1 Y_2)(t)$ from growing: the integrand in (56) is a vector which is approximately orthogonal to itself after a time scale $t \propto \mathcal{J}^{-1}N^{-1/2}$. As we will make

explicit in Section 5, this is a uniquely quantum mechanical effect relying on the time-independence of the Hamiltonian. It is absent in "classical" models of operator growth, such as the random unitary circuit.

In some respects, the phenomenon found above is of a similar flavor to the quantum Zeno effect [11], where a measured quantum state never decays. Here, the "measurement" is replaced by the fact that operators such as $Y_1 Y_2$ are "strongly coupled" (rotate rapidly into other operators), whereas $Z_1 Y_2$ slowly rotates into $Y_1 Y_2$. This hierarchy of rotation rates can mimic the quantum Zeno effect in a cartoon model. What we have found here is a many-body analogue of this "decoupling" which is more commonly studied in few state systems [11].

One shortcoming of the memory matrix result (54) is that we have not accounted for non-perturbative effects in $1/B$. Since $t_{\text{coh}} \propto \sqrt{N}$, these effects might be important and qualitatively change the physics on time scales $t \ll t_{\text{coh}}$, assuming $B \propto N^0$. We now more explicitly calculate $b(t)$, and argue that such non-perturbative effects do arise in the integrable model (43). The end result is that the actual coherence time is even larger than predicted in (55). Using the analytical exact diagonalization:

$$(X_2(t)|X_2) = 2^{-N} \text{tr}(X_2(t)X_2) = 2^{-N} \sum_{\mathbf{z},\mathbf{z}'} \langle z_1 z_2 \cdots z_N | e^{iHt} X_2 e^{-iHt} | z_1' z_2' \cdots z_N' \rangle \langle z_1' z_2' \cdots z_N' | X_2 | z_1 z_2 \cdots z_N \rangle$$

$$= 2^{-N} \sum_{\mathbf{z},\mathbf{z}'} \langle z_1 z_2 z_3 \cdots z_N | e^{iHt} X_2 e^{-iHt} | z_1 (-z_2) z_3 \cdots z_N \rangle$$

$$= \frac{1}{2^N} \sum_{\mathbf{z}} \left( \sum_{z_1} \langle z_1 | \left( \cos\left(t\sqrt{B^2 + B_{2+}^2}\right) + i\frac{BX_1 + B_{2+}Z_1}{\sqrt{B^2 + B_{2+}^2}} \sin\left(t\sqrt{B^2 + B_{2+}^2}\right) \right) \times \right.$$

$$\left. \left( \cos\left(t\sqrt{B^2 + B_{2-}^2}\right) - i\frac{BX_1 + B_{2-}Z_1}{\sqrt{B^2 + B_{2-}^2}} \sin\left(t\sqrt{B^2 + B_{2-}^2}\right) \right) |z_1\rangle \right)$$

$$= \frac{1}{2^{N-1}} \sum_{z_2 \cdots z_N} \left[ \cos(\omega_+ t)\cos(\omega_- t) + \frac{B^2 + B_{2+}B_{2-}}{\omega_+ \omega_-} \sin(\omega_+ t)\sin(\omega_- t) \right] \quad (58)$$

where

$$B_{2\pm}(z_3, \ldots, z_N) = \pm J_2 z_2 + \sum_{j=3}^{N} J_j z_j. \quad (59)$$

and $\omega_\pm = \sqrt{B^2 + B_{2\pm}^2}$. If $B = 0$, we can explicitly evaluate this expression:

$$(X_2(t)|X_2) = \frac{1}{2^{N-1}} \sum_{z_2 \cdots z_N} \left[ \cos(\omega_+ t)\cos(\omega_- t) + \text{sign}(B_+ B_-) \sin(\omega_+ t)\sin(\omega_- t) \right]$$

$$= \frac{1}{2^{N-1}} \sum_{z_2 \cdots z_N} \cos((B_+ - B_-)t) = \cos(2J_2 t), \quad (60)$$

in agreement with (33). When $B$ is small but non-zero, there is both a relative amplitude between the $\cos^2$ and $\sin^2$ terms, and a relative dephasing effect: $\text{sign}(B_+)\omega_+ - \text{sign}(B_-)\omega_- \neq J_2$. It is challenging to directly average over Gaussian random couplings analytically, but it is straightforward to evaluate (58) numerically for any $N$ and $B$, by "Monte Carlo" sampling over the disorder. The result is presented in Figure 6. It is clear that after a finite amount of time, $(X_2(t)|X_2)$ is not given by (54).

In the integrable model, we can explicitly track down the source of the problem by calculating the memory matrix kernel $\mathcal{K}(s) \approx (Y_1 Y_2(s)|Y_1 Y_2)$ a bit more carefully. The calculation is a direct extension

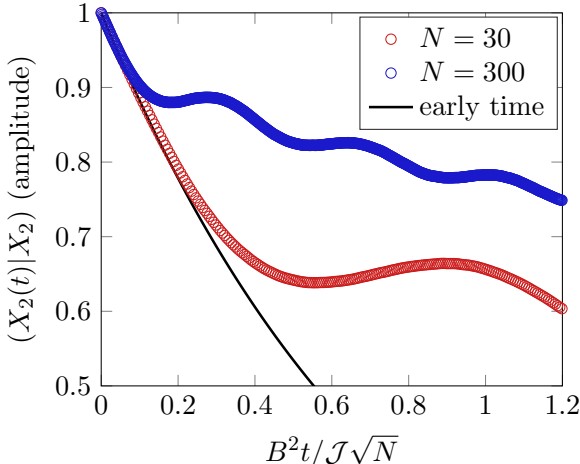

**Figure 6:** $(X_2(t)|X_2)$, as calculated from (58). To avoid spurious oscillations in the answer arising from $\cos(2J_2t)$, we fix $J_2 = \mathcal{J} = 1$ and only evaluate the correlation function at $t \in \pi\mathbb{Z}$. We set $B = 0.01$. The disorder average is evaluated numerically using $> 10^5$ samples, and statistical fluctuations in the answer are negligible. The solid black line is the prediction (54).

of (58) and we simply quote the result:

$$\mathcal{K}(t) = \frac{1}{2^{N-1}} \sum_{z_2 \cdots z_N} \left[ \cos(\omega_+ t) \cos(\omega_- t) + \frac{B^2 - B_{2+}B_{2-}}{\omega_+ \omega_-} \sin(\omega_+ t) \sin(\omega_- t) \right]. \tag{61}$$

The only difference is the relative minus sign in the last term, although this has a very large effect. If $B = 0$, this simply leads to $\mathbb{E}[\cos(2Jt)]$ which is indeed a Gaussian given by (50). However, when $B \neq 0$, there is an important discrepancy that arises. From the $\sin^2$ term in (61):

$$\frac{1}{2^{N-1}} \sum_{z_2 \cdots z_N} \frac{B^2}{\omega_+ \omega_-} \sin(\omega_+ t) \sin(\omega_- t)$$

$$= \frac{1}{2^N} \sum_{z_2 \cdots z_N} \frac{B^2}{\sqrt{(B^2 + B_{2+}^2)(B^2 + B_{2-}^2)}} \left( \cos((\omega_+ + \omega_-)t) + \cos((\omega_+ - \omega_-)t) \right)$$

$$\approx \cos(2J_2 t) \int_{-\infty}^{\infty} dJ \frac{e^{-J^2/N\mathcal{J}^2}}{\sqrt{2\pi N}\mathcal{J}} \frac{B^2}{\sqrt{(B^2 + (J+J_2)^2)(B^2 + (J-J_2)^2)}} \propto \cos(2J_2 t) \frac{B^2}{\mathcal{J}\sqrt{N}} \log\frac{|J_2|}{B}. \tag{62}$$

The last line is evaluated to leading order in $B$ and $N$, after disorder averaging over $J_3, \ldots, J_N$. The crucial point is that due to the overall $\cos(2J_2 t)$, this term in $\mathcal{K}(t)$ is in resonance with $(X_2(t)|X_2)$. The locality assumption required in (53) fails at times $t \gtrsim \mathcal{J}B^{-2}$ (up to logarithms). This is the time scale at which our numerical calculation of $(X_2(t)|X_2)$ disagrees with (54). We can also numerically evaluate (61) and check that these oscillations are present. As shown in Figure 7, at early times $\mathcal{K}(t)$ is well described by the Gaussian decay (even at small $N$), while at larger times an oscillatory factor arises at frequency $2J_2$.

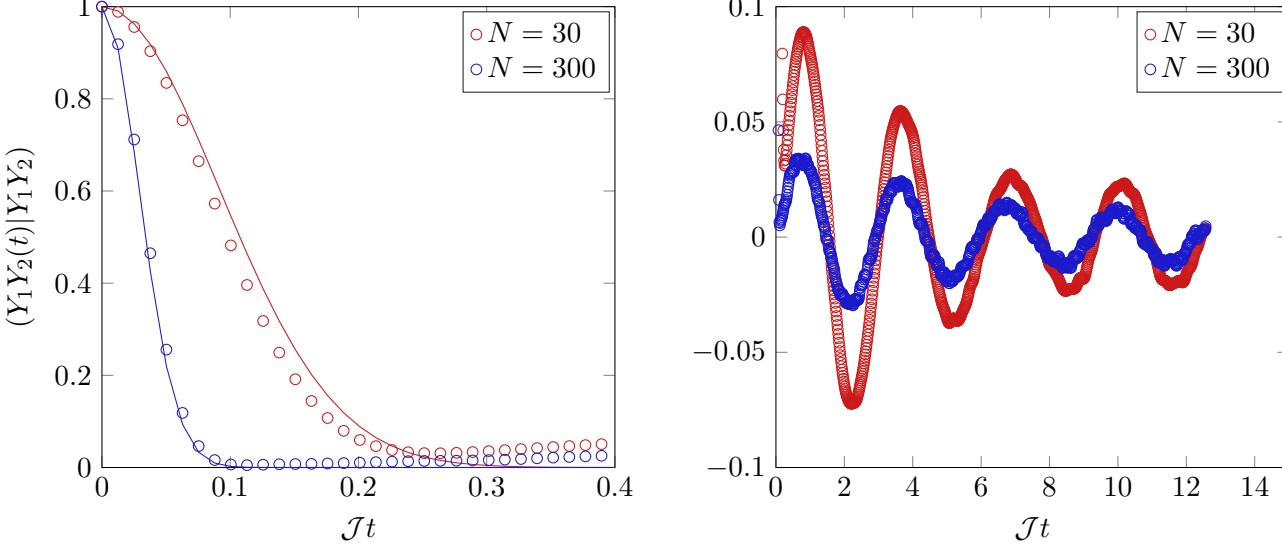

**Figure 7:** $(Y_1Y_2(t)|Y_1Y_2)$, as calculated from (61); we set $B = J_2 = \mathcal{J} = 1$. The disorder average is evaluated numerically using $> 10^5$ samples, and statistical fluctuations in the answer are negligible. Circles denote numerical data points. In the left panel, we focus on the early time limit where the Gaussian decay (parameter free theoretical prediction is the solid line) is observed. In the right panel, we observe the late time oscillations whose existence was argued for in the main text; the frequency of oscillations is $2J_2$, as predicted, and is independent of $N$.

## 4 Operator Growth without Integrability

We now turn to operator growth in a generic and chaotic model on the star graph. Unlike before, it is now possible for all operators to grow large in constant time (at infinite temperature).

### 4.1 Decay of Two Point Functions

To justify this, we study the early time dynamics of the operator $X_1(t)$. For simplicity, we assume that $B_i^\alpha = 0$ in the argument that follows; in the thermodynamic limit $N \to \infty$ this is acceptable (at early times). We will first calculate

$$a_1(t) = 2^{-N}\mathrm{tr}(Z_1(t)Z_1) = (Z_1|\mathrm{e}^{\mathcal{L}t}|Z_1), \tag{63}$$

though from this calculation it is possible to obtain further information as well. A useful "lemma" is the following: if $A$ and $B$ are tensor products of Pauli matrices, then so are $AB$ and $BA$. Moreover, one always finds that $[A, B] = \eta AB$ where $\eta$ is a constant O(1) prefactor. With this in mind, we now study

$$(Z_1|\mathrm{e}^{\mathcal{L}t}|Z_1) = \sum_{n=0}^{\infty} \frac{t^n}{n!}(Z_1| \left( \sum_{j\alpha\beta} \mathcal{L}_{1j}^{\alpha\beta} \right)^n |Z_1). \tag{64}$$

where $\mathcal{L}_{1j}^{\alpha\beta} = \mathrm{i}J_j^{\alpha\beta}[X_1^\alpha X_j^\beta, \circ]$. In order for the inner product in (64) to be non-vanishing, the product of $\mathcal{L}_{1j}^{\alpha\beta}$ must return $X_1$ back to itself: there can be no additional Paulis on sites $2, \ldots, N$. At leading

order at large $N$, this implies that only even powers in $n$ contribute to the above sum, and that at order $n = 2k$, there are $O(N^k)$ different terms in the sum to consider, consisting of all possible pairs of couplings $\{J_{j\ell}^{\alpha\beta\ell}, J_{j\ell}^{\alpha'\beta\ell}\}$ for $\ell = 1, \ldots, k$. This is a dramatic reduction over the $O(N^{2k})$ terms which were initially present. For simplicity, we will further disorder average over $J_j^{\alpha\beta}$, which enforces $\alpha = \alpha'$ in our pairs of couplings (note that this is not a significant reduction in the number of terms to consider).

In the large $N$ limit, each site $j$ will almost surely show up exactly once until the sum above reaches order $k \propto \sqrt{N}$. This result follows from a simple combinatoric argument which is found in [23]. For us, this leads to an enormous simplification whenever we are interested in terms in (64) at orders $n \lesssim \sqrt{N}$: we can treat the Paulis on sites $j = 2, \ldots, N$ as independent, identically distributed *classical random variables* obeying

$$\mathbb{P}\left(X_j^\alpha = 1\right) = \mathbb{P}\left(X_j^\alpha = -1\right) = \frac{1}{2}. \tag{65}$$

Since no two distinct Paulis can ever show up on sites $j = 2, \ldots, N$ until the same site is chosen twice, at early times all Paulis on sites $j = 2, \ldots, N$ commute with each other and are thus "classical". Averaging over the $X_j^\alpha$ precisely encodes the requirement above that each Pauli on sites $j = 2, \ldots, N$ must show up twice in $a_1(t)$, at leading order in $N$. To estimate the time at which terms of order $k \propto \sqrt{N}$ become important, we ask when

$$1 \lesssim \left( \begin{array}{c} N \\ k \end{array} \right) \frac{(\mathcal{J}t)^{2k}}{(2k)!} \sim \exp\left[ 2k \log \frac{\sqrt{N}\mathcal{J}t}{2k} \right].$$

We conclude that when $\mathcal{J}t \ll 1$, terms of order $k \gtrsim \sqrt{N}$ are exponentially suppressed and can be neglected.

Thus we have found an enormous simplification: $a_1(t)$ is quantitatively captured by the "disorder averaged" dynamics of a *two level system* when $\mathcal{J}t \ll 1$. The Hamiltonian $H_2$ of the two level system is

$$H_2 \approx \sum_{\alpha=1}^{3} h_\alpha X_1^\alpha, \tag{66}$$

where

$$h^\alpha = \sum_{\beta=1}^{3} \sum_{j=2}^{N} J_j^{\alpha\beta} X_j^\beta, \tag{67}$$

where $X_j^\beta$ are now classical random variables to be averaged over. We must now evaluate

$$a_1(t) = \mathbb{E}\left[ \text{tr}\left( e^{iH_2 t} Z_1 e^{-iH_2 t} Z_1 \right) \right] \tag{68}$$

where $\mathbb{E}[\cdots]$ denotes the disorder average over (65). It is straightforward to analyze the trace before disorder averaging:

$$a_1(t) = \mathbb{E}\left[ 1 - 2\sin^2\left( \sqrt{h_X^2 + h_Y^2 + h_Z^2} \, t \right) \frac{h_X^2 + h_Y^2}{h_X^2 + h_Y^2 + h_Z^2} \right]. \tag{69}$$

We emphasize again that the disorder average here is exact and is the needed prescription to convert from the two level problem back to the many-body problem on the star graph.

There is no need to individually average over $X_j^\beta$ in (69). Using the central limit theorem, the probability density functions of $h_X$, $h_Y$ and $h_Z$ are all equal and given by, e.g.

$$\text{p}(h_X)\text{d}h_X = \frac{e^{-h_X^2/6N\mathcal{J}^2}}{\sqrt{6\pi N \mathcal{J}}}\text{d}h_X. \tag{70}$$

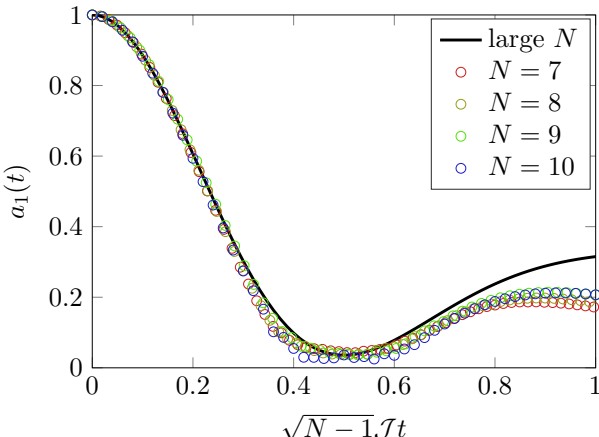

**Figure 8:** Universal early time dynamics in $a_1(t)$ is well described by the prediction (71) (solid line) of the effective two level system, even in numerical simulations on relatively small system sizes (circles). The color of the markers denotes the value of $N$, as given in the legend. At later times $\mathcal{J}t \sim 1$, the two level model breaks down and $a_1(t)$ will ultimately decay to 0.

The disorder average is actually easiest to do by going to spherical coordinates and disorder averaging over a three-dimensional Gaussian distribution:

$$
\begin{aligned}
a_1(t) &= 2\pi \int\limits_0^\infty \mathrm{d}h \int\limits_0^\pi \mathrm{d}\theta \, h^2 \sin\theta \frac{\mathrm{e}^{-h^2/6N\mathcal{J}^2}}{(6\pi N)^{3/2}\mathcal{J}^3} \left(1 - 2\sin^2\theta \sin^2(ht)\right) \\
&= \frac{1}{3} + \frac{2}{3}\mathrm{e}^{-6N\mathcal{J}^2 t^2} \left(1 - 12N\mathcal{J}^2 t^2\right).
\end{aligned}
\tag{71}
$$

Polar coordinates are oriented such that $h_Z = h\cos\theta$. This formula holds whenever $\mathcal{J}t \ll 1$. Remarkably, (71) predicts that $\mathrm{tr}(Z_1(t)Z_1)$ does *not* decay on the time scale $t \sim N^{-1/2}$, as the non-trivial correlator $\mathrm{tr}(X_1(t)X_1)$ does in the Ising model. The reason is deceptively simple in the effective two-level system: around a "third" of the effective field felt by the central spin points in the $Z$ direction, and will not decay the operator $Z_1$. From the many-body perspective, this effect is much more remarkable. Figure 8 compares our theoretical prediction to numerical simulations for early times. We see excellent agreement between theory and numerics at early times, along with clear evidence for the predicted *minimum* in $a_1(t)$ at a time $\mathcal{J}t \sim N^{-1/2}$. Due to finite size effects, we are unable to see the prolonged saturation of $a_1(t)$ for intermediate times.

(71) also leads to another remarkable observation. Consider the decay of the operator $Z_2$, as measured by the two point function

$$
a_2(t) = (Z_2(t)|Z_2).
\tag{72}
$$

We can approximately evaluate this using the memory matrix formalism, as in Section 3.3. Again neglecting the 1-local $B_i^\alpha$ terms in (7), and using that in the large $N$ limit $X_1^\alpha$ and $X_1^\alpha X_2^\beta$ have essentially identical dynamics, we conclude that

$$
\frac{\mathrm{d}}{\mathrm{d}t}a_2(t) \approx -24\mathcal{J}^2 \int\limits_0^t \mathrm{d}s\, a_1(t-s)a_2(s).
\tag{73}
$$

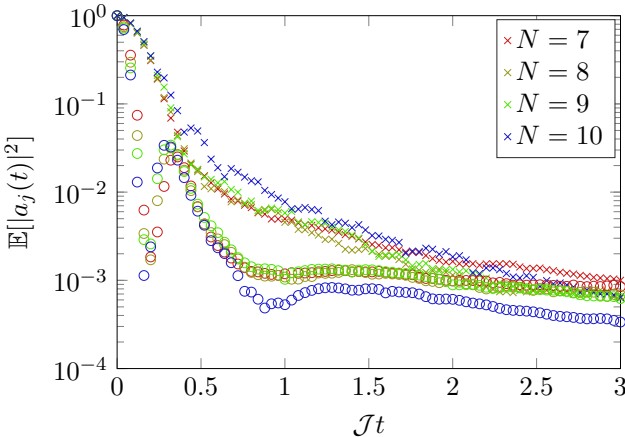

**Figure 9:** The decay of $a_j(t)$ for $j = 1$ (circles) and $j = 2$ (crosses) is essentially independent of $N$ once $\mathcal{J}t \gtrsim 1$. The color of the markers denotes the value of $N$, as given in the legend.

Upon disorder averaging, this expression is exact to leading order in $1/N$. The function $a_1(t) \approx \frac{1}{3}$ for $N^{-1/2} \lesssim \mathcal{J}t \lesssim 1$. Therefore, unlike in (53), the memory function $a_1(t-s)$ does *not* have a vanishingly small integral in the large $N$ limit. $a_2(t)$ will not have a parametrically long coherence time as in (55). Instead, we expect both $a_1(t)$ and $a_2(t)$ to decay to zero on the time scale $\mathcal{J}^{-1}$ (likely exponentially quickly). Figure 9 confirms that in our numerics, $a_1(t)$ and $a_2(t)$ decay on similar, $N$-independent time scales once $\mathcal{J}t \gtrsim 1$. One interesting observation in our numerics is that $a_1(t)$ and $a_2(t)$ do not appear to decay to the value $2^{-N}$ (at which point $a_1(t)$ has become "completely random"). It is unclear whether this is a finite size effect.

## 4.2 | Out of Time Ordered Correlators

Further information about growing operators is obtained by studying OTOCs: at infinite temperature, we evaluate

$$C_{jk}(t) = 2^{-N} \left| \text{tr} \left( [Z_j(t), Z_k]^2 \right) \right|. \tag{74}$$

At early times, we can evaluate $C_{12}(t)$ analytically using the effective two level system (66):

$$C_{jk}(t) \approx 2 \times \text{tr} \left( \left( \frac{\partial Z_1(t)}{\partial X_k} \right)^2 + \left( \frac{\partial Z_1(t)}{\partial Y_k} \right)^2 \right). \tag{75}$$

Using that (at early times)

$$Z_1(t) = \sum_{\alpha=1}^{3} c_\alpha(t) X_1^\alpha \tag{76}$$

with

$$c_X(t) = \cos(ht)\sin(ht)\frac{h_Y}{h} + \sin^2(ht)\frac{h_X h_Z}{h^2}, \tag{77a}$$

$$c_Y(t) = -\cos(ht)\sin(ht)\frac{h_X}{h} + \sin^2(ht)\frac{h_Y h_Z}{h^2}, \tag{77b}$$

$$c_Z(t) = \cos^2(ht) + \sin^2(ht)\frac{h_Z^2 - h_X^2 - h_Y^2}{h^2}, \tag{77c}$$

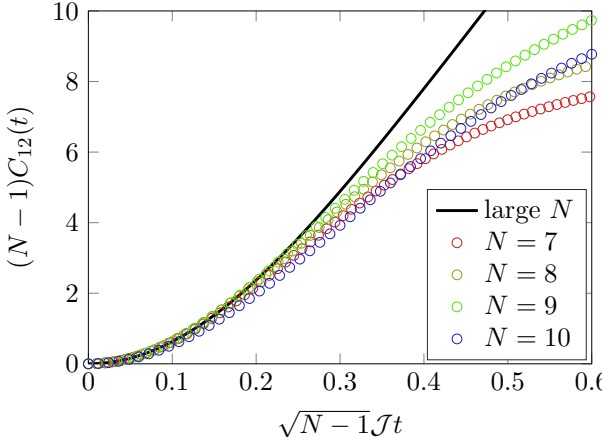

**Figure 10:** The growth of $C_{12}(t)$ is universal for early times: we compare the theoretical prediction (79) to numerical simulations.

where $h = \sqrt{h_X^2 + h_Y^2 + h_Z^2}$, we find that after a bit of algebra,

$$C_{12}(t) = 8\mathcal{J}^2 \times \mathbb{E}\left[\frac{3 + 2h^2t^2 + (1 - 2h^2t^2)\cos(2\theta) - (3 + \cos(2\theta))\cos(2ht)}{h^2}\right] \tag{78}$$

where we have again used the polar representation of $h_\alpha$. This can be analytically evaluated as before; the result is

$$C_{12}(t) = \frac{64}{9N}\left[1 + 3N\mathcal{J}^2t^2 - e^{-6N\mathcal{J}^2t^2}\right]. \tag{79}$$

Our numerics confirms this behavior at early times: see Figure 10.

We can also study the behavior of OTOCs on longer time scales $\mathcal{J}t \gtrsim 1$. Figure 11 plots both $C_{12}(t)$ and $C_{23}(t)$ as a function of $\mathcal{J}t$ – we observe that both become large after a finite, $N$-independent time. This is more compelling evidence for our claim that in the chaotic models on the star graph, quantum information is not protected on nodes $j = 2, \ldots, N$ for a long time. In a fully scrambled system, we would expect that $C_{jk}(\infty) = 2$. Both $C_{12}(t)$ and $C_{23}(t)$ appear to grow quite close to 2 quickly, while the saturation is much slower.

At finite temperature, we can also compute the OTOC

$$C_{jk}(t) = \left|\frac{\operatorname{tr}\left(\sqrt{\rho}[Z_j(t), Z_k]\sqrt{\rho}[Z_j(t), Z_k]\right)}{\operatorname{tr}\left(\sqrt{\rho}Z_j\sqrt{\rho}Z_j\right)\operatorname{tr}\left(\sqrt{\rho}Z_k\sqrt{\rho}Z_k\right)}\right|. \tag{80}$$

The results are also plotted in Figure 11. At both $\beta\mathcal{J} = 0$ and 0.1, it appears that our numerics have approximately converged and that generic OTOCs grow large in constant time.

At lower temperatures, the typical value of the OTOC at later times becomes increasingly small. Figure 12 plots the median value of $|C_{2j}(t)|$ in numerical simulations at much lower temperatures at a fixed, later value of time. We see that for sufficiently high temperature $\beta\mathcal{J} \lesssim 0.15$ the numerics appear to converge, while at higher temperatures it is plausible that the OTOC vanishes in the $N \to \infty$ limit. Hence, while we cannot definitively rule out the possibility that our model enters a phase at any $\beta\mathcal{J} > 0$ where typical OTOCs do not grow large at times $t \lesssim \beta \log N$, we believe that Figure 12 provides substantial evidence that the chaos bound does not hold at sufficiently high temperature (if it holds at any temperature). We are also unsure whether the question of chaos bound violation is linked to the

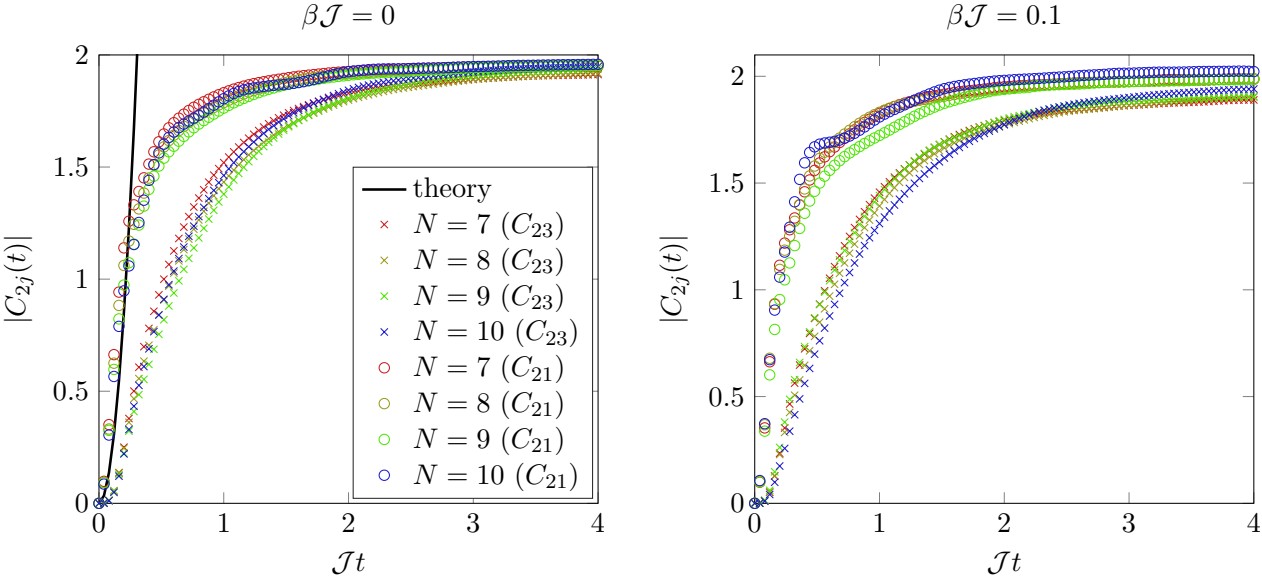

**Figure 11:** The late time dynamics of $C_{2j}(t)$ is universal at high temperature, and implies that all operators grow large in constant time in the thermodynamic limit.

existence of the "glassy" phase or not. While we cannot accurately estimate decay rates of operators in our numerics, this model could also violate the postulate of [21] that no $k$-local quantum system can have all small operators decay (in some suitable sense) before the Planckian time scale $\hbar/k_\mathrm{B}T$.

We found in our numerics that there are enormous sample-to-sample fluctuations in the value of $C_{2j}(t)$ once $\beta\mathcal{J} \gtrsim 1$ – we believe these are a chaotic analogue of the $\cosh^2(\beta J_2)$ in (42). It could be interesting to understand these fluctuations further. This is further evidence for the "glassy" nature of the low temperature phase of this model.

## 5  Stochastic Quantum Dynamics

### 5.1  Brownian Hamiltonian Dynamics

In this section, we describe the qualitatively different dynamics which arise when we instead consider a Brownian Hamiltonian [7, 8]

$$H(t) = \sum_{j=2}^{N} \sum_{\alpha,\beta=1}^{3} J_j^{\alpha\beta}(t) X_1^\alpha X_j^\beta \tag{81}$$

where the time dependent couplings are Gaussian white noise:

$$\mathbb{E}\left[ J_j^{\alpha\beta}(t) J_{j'}^{\alpha'\beta'}(t') \right] = \frac{\mathcal{J}}{4} \delta_{jj'} \delta^{\alpha\alpha'} \delta^{\beta\beta'} \delta(t-t'). \tag{82}$$

For convenience, we have ignored the possibility of 1-local terms, which will not change our results, given the ansatz (82). Time evolution of operators with this Hamiltonian is as usual: $\partial_t|\mathcal{O}) = \mathcal{L}(t)|\mathcal{O}) = |\mathrm{i}[H(t),\mathcal{O}])$.

The easiest way to understand the dynamics here is to think about an "operator mixed state"

$$|\rho] \equiv |\mathcal{O})(\mathcal{O}| \equiv |\mathcal{O}\otimes\mathcal{O}]. \tag{83}$$

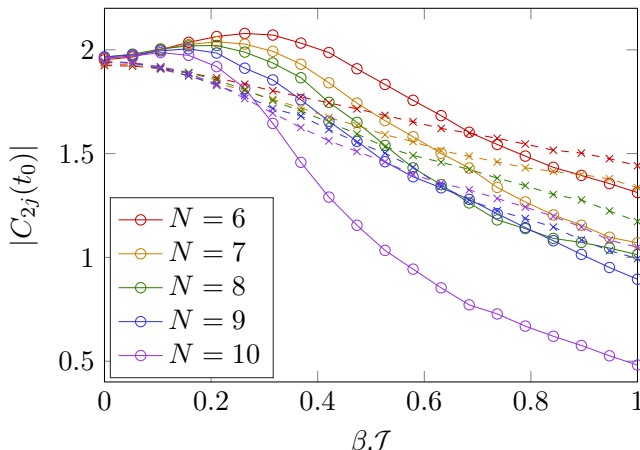

**Figure 12:** The temperature dependence of the median value of $C_{2j}(t_0)$ at the fixed time $t = 4/\mathcal{J}$, averaged over random instances of (7). For sufficiently small $\beta\mathcal{J} \lesssim 0.15$ it appears that the result is independent of $N$.

Time evolution of $|\rho]$ is generated by

$$\partial_t |\rho] \equiv \mathcal{M}|\rho] = (\mathcal{L} \otimes 1 - 1 \otimes \mathcal{L})|\rho]. \tag{84}$$

The reason this doubling of operators is so useful is as follows. $|\rho]$ is a density matrix whose diagonal components (in any basis) encode the "probability" that an operator is in a given state. If we wish to compute infinite temperature OTOCs such as $\mathrm{tr}([X_1(t), X_2]^2)$, it suffices to calculate the probability that $X_1(t)$ is in a "state" with either a $Y_2$ or a $Z_2$. We also note that since energy is not conserved due to time dependence of $H(t)$, the natural ensemble of interest is the infinite temperature ensemble.

Let us consider the basis of operators generated by tensor products of $\{1_i, X_i, Y_i, Z_i\}$ for each $i$. Remarkably, in this basis, if $|\rho(0)]$ is diagonal at time $t = 0$, $\mathbb{E}[|\rho(t)]]$ will remain diagonal for all time $t$. To show this, we explicitly evaluate

$$\mathbb{E}\left[e^{\mathcal{M}t}\right] = \mathbb{E}\left[\sum_{n=0}^{\infty} \int_0^t \mathrm{d}t_1 \int_0^{t_1} \mathrm{d}t_2 \cdots \int_0^{t_{n-1}} \mathrm{d}t_n \, \mathcal{M}(t_1)\mathcal{M}(t_2)\cdots\mathcal{M}(t_n)\right]$$

$$= \sum_{n=0}^{\infty} \int_0^t \mathrm{d}t_1 \int_0^{t_1} \mathrm{d}t_2 \cdots \int_0^{t_{2n-1}} \mathrm{d}t_{2n} \, \mathbb{E}\left[\mathcal{M}(t_1)\mathcal{M}(t_2)\right]\cdots\mathbb{E}\left[\mathcal{M}(t_{2n-1})\mathcal{M}(t_{2n})\right] \tag{85}$$

To understand the second line, observe that there must be an even number of $\mathcal{M}$ because of the Gaussian random variables sitting in $H(t)$, and therefore $\mathcal{L}(t)$ and $\mathcal{M}(t)$. Moreover, due to the white noise (82), we find that $\mathbb{E}[\mathcal{M}(t)\mathcal{M}(t')] \propto \delta(t - t')$. So $t_1 = t_2$, $t_3 = t_4$, etc. If, for example, we had averaged $\mathcal{M}(t_1)$ with $\mathcal{M}(t_3)$ and $\mathcal{M}(t_2)$ with $\mathcal{M}(t_4)$, we would find $t_1 = t_2 = t_3 = t_4$. While the integrals over $t_3$ and $t_4$ in the second line of (85) are removed by the $\delta$ function in (82), there remains a residual factor of $\int_{t_1}^{t_1} \mathrm{d}t_2 = 0$. Hence, we cannot average any $\mathcal{M}(t)$ out of order.

Let us now explicitly evaluate

$$\mathcal{W} \equiv \int_0^t \mathrm{d}s \, \mathbb{E}\left[\mathcal{M}(t)\mathcal{M}(s)\right] = \int_0^t \mathrm{d}s \, \mathbb{E}\left[\mathcal{L}(t)\mathcal{L}(s) \otimes 1 + 1 \otimes \mathcal{L}(t)\mathcal{L}(s) \otimes -\mathcal{L}(t) \otimes \mathcal{L}(s) - \mathcal{L}(s) \otimes \mathcal{L}(t)\right]. \tag{86}$$

Define $\mathcal{L}_j^{\alpha\beta}(t) = \mathrm{i}[J_j^{\alpha\beta}(t)X_1^\alpha X_j^\beta, \circ]$, and define the diagonal states

$$|X_1^{a_1} X_2^{a_2} \cdots X_N^{a_N}] \equiv |X_1^{a_1} X_2^{a_2} \cdots X_N^{a_N} \otimes X_1^{a_1} X_2^{a_2} \cdots X_N^{a_N}]. \tag{87}$$

Here $a = 0, 1, 2, 3$ runs over the four basis operators on each site, and $a = 0$ denotes the identity. Then using (81) and (82), we find that

$$\mathcal{W}|X_i^{a_i}] = \mathcal{J} \sum_{a',b',\alpha,\beta,j} 4K_{\alpha,\beta}^{a_1 a_j, a'b'} \left( -2|X_1^{a_1} \cdots X_j^{a_j} \cdots] + 2|X_1^{a'} \cdots X_j^{b'} \cdots] \right) \tag{88}$$

where the symmetric matrix $K_{\alpha\beta}^{ab,a'b'} = K_{\alpha\beta}^{a'b',ab}$ has entries:

$$K_{\eta\theta}^{00,00} = K_{\eta\theta}^{00,0\alpha} = K_{\eta\theta}^{0\alpha,\beta0} = K_{\eta\theta}^{0\alpha,0\beta} = K_{\eta\theta}^{\alpha0,\beta0} = K_{\eta\theta}^{\alpha\beta,00} = K_{\eta\theta}^{\alpha\beta,\gamma\delta} = 0, \tag{89a}$$

$$K_{\eta\gamma}^{0\alpha,\delta\beta} = K_{\gamma\eta}^{\alpha0,\beta\delta} = \delta_{\delta,\eta} |\epsilon^{\alpha\beta\gamma}|. \tag{89b}$$

In the above formula, all Greek letters denote distinct indices. Note that there are a number of cancelling numerical prefactors in (88). A factor of $\frac{1}{2}$ comes from considering any symmetric smoothing of $\delta(t)$ (e.g. $\lim_{\epsilon\to 0}(2\epsilon)^{-1}\mathrm{e}^{-|t|/\epsilon}$), and noting that the integral in (86) only covers "half" of the smoothed function. A factor of 4 comes from the doubled commutators in $\mathcal{L}\otimes\mathcal{L}$ or $\mathcal{L}^2\otimes 1$, and the factor of 2 in all non-vanishing Pauli commutators: e.g., $\mathrm{i}[X_1 X_2, Y_1] = -2Z_1 X_2$. Note that the only non-vanishing Pauli commutators involve either growing or shrinking the operator by one Pauli.

The key observation from (88) is that if $|\rho(0)]$ is diagonal at $t = 0$,

$$\mathbb{E}\left[|\rho(t)]\right] = \mathrm{e}^{\mathcal{W}t}|\rho(0)] \tag{90}$$

is also diagonal. This is because $\mathcal{W}$ is a negative semidefinite matrix which generates a continuous time Markov process on the *classical* state space of all $4^N$ strings of Paulis/identities. As far as average operator growth is concerned, therefore, the Brownian quantum dynamics is governed by a *classical stochastic process* with no quantum fluctuations. Due to the high symmetry of our problem, we need not study the probability of finding each string of Paulis/identities separately. We may instead calculate the probability $p_m(t)$ that we have *any* Pauli on vertex 1 *and any* Pauli on $m$ other vertices, and the probability $q_m(t)$ that we have identity on vertex 1 and any Pauli on $m$ other vertices. By symmetry, at all times the distribution within these $2N$ sectors we have identified is uniform. The Markov process (90) can, with a little extra work, be simplified to

$$\mathcal{J}^{-1}\partial_t p_m = 6mq_m + 6(N - m)p_{m-1} + 2(m + 1)p_{m+1} - (6(N - m - 1) + 4m)p_m, \tag{91a}$$

$$\mathcal{J}^{-1}\partial_t q_m = 2mp_m - 6mq_m. \tag{91b}$$

To analyze (91), observe that we are interested in the large $N$ limit, where we expect that $p_m$ and $q_m$ are (at times $\mathcal{J}t \gg N^{-1}$) smooth functions of $m$. As such, we rescale $m$ to

$$Q = \frac{m}{N}. \tag{92}$$

and approximate (91) with

$$\mathcal{J}^{-1}\partial_t p \approx 2NQ(3q - p) - (6 - 8Q)\partial_Q p + \frac{6 - 4Q}{N}\partial_Q^2 p, \tag{93a}$$

$$\mathcal{J}^{-1}\partial_t q \approx 2NQ(p - 3q). \tag{93b}$$

These equations suggest that once $Q \gg N^{-1/2}$, we should treat these equations order by order in $N^{-1}$. At leading order, we demand that $p = 3q$. In other words, $p \approx \frac{3}{4}P$, where $P(m)$ is the probability of finding $m$ Paulis on the vertices $2, \ldots, N$. At next order, we find

$$\partial_t P \approx -\frac{3}{4}(6 - 8Q)\partial_Q P \approx -\partial_Q \left(\frac{3}{4}(6 - 8Q)P\right), \tag{94}$$

which is a transport equation whose solution is $P(Q, t) \approx \delta(Q - \langle Q(t)\rangle)$, with

$$\langle Q(t)\rangle \approx \frac{3}{4}\left(1 - e^{-6\mathcal{J}t}\right). \tag{95}$$

At second derivative order, we find small $\frac{1}{N}$-suppressed fluctuations which will only be important once $\langle Q(t)\rangle$ saturates to its final value of $\frac{3}{4}$. Note that we may safely perform the second approximation in (94) because moving the $\partial_Q$ through $6 - 8Q$ only causes a O$(1/N)$ correction to the leading order ($\partial_Q^0$) term in (93).

Hence, we obtain a universal picture of operator growth on the star graph under the Brownian Hamiltonian model (81). At early times, the operator will (easily) fluctuate onto the central vertex 1. After a short wait of $\Delta t \propto N^{-1/2}$, any operator on the central vertex begins to grow deterministically. The fraction of vertices with a non-trivial Pauli obeys (95), with no residual statistical fluctuations in the thermodynamic limit $N = \infty$. After a finite time $t \sim \mathcal{J}^{-1}$, an initially simple operator acts on O$(N)$ sites. The fast scrambling conjecture does not apply to operator growth in this model.

One useful way to visualize operator growth in this model is to not solve the master equation (90) for the Markov process, but to instead study instances of the equivalent stochastic process. This is presented in Figure 13. As argued on general grounds above, statistical fluctuations become negligible as $N$ grows larger. The only non-universality in the dynamics at $N = \infty$ results from the fact that if the initial small operator does not act on vertex 1, there is a random and finite waiting time before the operator grows on to vertex 1.

The Brownian dynamics described here is completely different from the actual quantum dynamics generated by a time-independent Hamiltonian. The origin of this discrepancy boils down to the fact that the Brownian dynamics is equivalent to a classical Markov chain, in which probabilities for finding operators in different "states" add linearly. However, when evolving operators under a time-independent Hamiltonian, these probabilities add quadratically. In the Brownian dynamics, the probability $p_1$ for the operator to exist exclusively on the central site is negligible after a finite time evolution, while this probability is finite in the chaotic quantum dynamics. In fact, in the (nearly) integrable models where $p_1$ does become negligible for $\mathcal{J}t \gtrsim N^{-1/2}$, the rapid decay of the operator $X_1$ prevents the growth of other operators such as $X_2$.

## 5.2 ▌ Random Unitary Circuit

As a final note, we also comment that the quantum Brownian dynamics on the star graph is also qualitatively different from random unitary circuit dynamics on the star graph [4] at early times.[1] To ensure a proper comparison with the Brownian model above, we take the local Hilbert space dimension to be 2 for each vertex. The random unitary circuit dynamics on the star graph proceed as follows: at time steps $\Delta t = (\mathcal{J}N)^{-1}$, evolve operators as $\mathcal{O}(t + \Delta t) = U_t^\dagger \mathcal{O}(t)U_t$, where $U_t$ is a "small" unitary operator, randomly chosen as follows: first, choose a vertex $j \in \{2, \ldots, N\}$ uniformly at random (i.e., choose a random edge from the star graph). Next, choose the unitary matrix $U_t = U_{1j} \bigotimes_{k \neq 1, j} 1_k$ with $U_{1j}$ a randomly chosen matrix from the Haar ensemble on U(4), acting on vertices 1 and $j$. The growth of operators in this random unitary circuit maps on to a Markov chain, analogous to a discrete time version of (90). The

---

[1] As $t \to \infty$, both quantum dynamical systems fully scramble any operator: see e.g. [24].

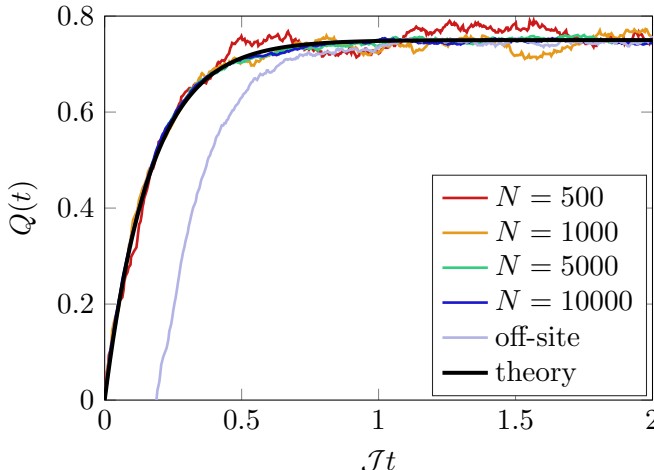

**Figure 13:** Comparison of the theoretical prediction (95), shown as the black line, for $Q(t)$ to numerical simulations of the associated stochastic process. While only a single instance is shown, there are not significant fluctuations between instances (at large $N$). Simulations for which the initial operator is on the central vertex 1 are denoted in red/yellow/green/blue; the value of $N$ is shown in the legend. Independently of $N$, all data collapses onto a universal curve. We also show a single instance of a process where the initial operator is on vertex 2 (labeled "off-site"): once the operator spreads to vertex 1 the dynamics follows the universal growth pattern (95).

state space of this Markov chain can, using symmetries, be made identical to the state space of (91): we need only keep track of whether the operator exists on vertex 1, along with how many remaining sites $m$ have a non-trivial operator. Let us call these states $0m$ and $1m$, with the $0/1$ denoting vertex 1 being unoccupied/occupied. The transition rates for this Markov chain are:

$$\mathbb{P}[0m \to 1m] = \frac{3}{5}\frac{m}{N-1}, \tag{96a}$$

$$\mathbb{P}[0m \to 1(m-1)] = \frac{1}{5}\frac{m}{N-1}, \tag{96b}$$

$$\mathbb{P}[1m \to 0m] = \frac{1}{5}\frac{m}{N-1}, \tag{96c}$$

$$\mathbb{P}[1m \to 1(m-1)] = \frac{1}{5}\frac{m}{N-1}, \tag{96d}$$

$$\mathbb{P}[1m \to 0(m+1)] = \frac{1}{5}\frac{N-1-m}{N-1}, \tag{96e}$$

$$\mathbb{P}[1m \to 1(m+1)] = \frac{3}{5}\frac{N-1-m}{N-1}. \tag{96f}$$

Transition rates not shown correspond to states not changing during each discrete time step, and are easily computed by demanding that the probability $\sum_j \mathbb{P}[xi \to 0j] + \mathbb{P}[xi \to 1j] = 1$.

In this random unitary circuit, we may again calculate $\langle Q(t) \rangle$. Once $m \gg 1$, we can approximate the discrete Markov chain (96) by a continuum Fokker-Planck equation, analogous to (91):

$$\partial_t p \approx \mathcal{J}N\left[\frac{4}{5}Qq - \frac{p}{5} + \frac{Q}{5N}\partial_Q q - \frac{3-4Q}{5N}\partial_Q p + \frac{Q}{10N^2}\partial_Q^2 q + \frac{3-2Q}{10N^2}\partial_Q^2 p\right], \tag{97a}$$

$$\partial_t q \approx \mathcal{J}N\left[\frac{p}{5} - \frac{4}{5}Qq - \frac{1-Q}{5N}\partial_Q p + \frac{1-Q}{10N^2}\partial_Q^2 p\right]. \tag{97b}$$

The method of approximate solution is similar to before. At leading order in $N$, (97) implies that

$$p = P - q = \frac{4Q}{4Q+1}P. \tag{98}$$

At next to leading order in $N$, we obtain an approximate transport equation

$$\partial_t P \approx -\mathcal{J}\partial_Q\left(\frac{Q(3-4Q)}{4Q+1}P\right). \tag{99}$$

after neglecting $\mathrm{O}(1/N)$ corrections to (98) which follow from moving derivatives in $Q$ through non-constant prefactors. Hence,

$$\frac{\mathrm{d}}{\mathrm{d}t}\langle Q\rangle \approx \mathcal{J}\langle Q\rangle\frac{3-4\langle Q\rangle}{1+4\langle Q\rangle}. \tag{100}$$

We do not find an elegant closed form solution to this equation, but it is straightforward to solve numerically for all time. We can also observe that at early times,

$$\langle Q(t)\rangle \approx \frac{1}{N}\mathrm{e}^{3\mathcal{J}t}, \tag{101}$$

which means that, unlike in either the time-independent Hamiltonian dynamics or the Brownian dynamics, the size of no operator grows faster than exponentially in this random unitary circuit (see also [4]). A comparison of our numerical solution to individual instances of the random unitary circuit is presented in Figure 14. Upon shifting the time variable in individual instances, to account for non-negligible early time fluctuations in the size of the operator (a consequence of the transition (96e), which was negligible in the Brownian dynamics at early times), we find excellent agreement between simulations and (100).

## 6 Outlook

In this paper, we have explored the 2-local quantum model (7) on the star graph on $N$ vertices. We have argued that in the thermodynamic limit $N \to \infty$, this model can "scramble" information (as measured by the growth of OTOCs) in constant time, even though interactions are few-body. Numerics suggests that this constant time saturation of OTOCs persists to finite temperature, implying the first counterexample to the bound (5) in a chaotic 2-local many-body system. As such, an intuitive understanding of the fast scrambling bound (1) as a limitation on chaos and operator growth does not appear to be correct. We propose that, as in [4], fast scrambling is better understood as a constraint on the generation of entanglement. As $(X_2(t)|X_2)$ appears to never decay faster than exponentially in our model, (1) is a constraint on the time it takes to create (nearly) maximal entanglement [4].

Despite the extremely rapid growth of operators in this model, we emphasize that operator growth is *quantum* in nature. There are significant quantitative differences between how operators grow in quantum Hamiltonian evolution versus stochastic models (Brownian dynamics or random unitary circuits). In particular, operator growth in the chaotic model appears to be so fast because of a curious coexistence of operator growth and quantum coherence. In the quantum random Ising model, operator growth on the central vertex is also very fast (in the thermodynamic limit), yet the rapid growth of operators on the central vertex protects the remaining qubits from decoherence. We conjecture that the qualitative features of operator growth on the star graph persist in studies of quantum models on more general heterogeneous networks such as scale-free networks [25]. These networks also have highly connected nodes, analogous to

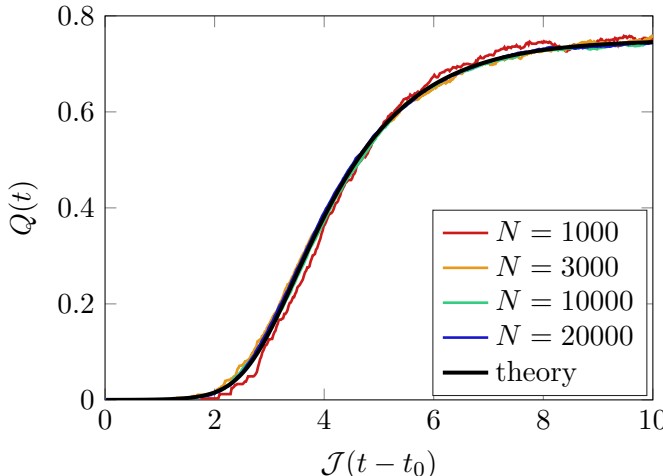

**Figure 14:** Comparison of the theoretical prediction (100), shown as the black line, for $Q(t)$ to numerical simulations of the associated random unitary circuit. We show single instances of the circuit for each value of $N$, and always assume that the initial operator is on the central vertex 1. The time it takes for the operator to begin its exponential growth is itself a random variable; after shifting by an appropriate constant $t_0$ depending on the particular realization, all data collapses onto a universal curve. Stochastic fluctuations become suppressed at large $N$.

the central ($j = 1$) vertex of the star graph, albeit to a much lesser extent: the maximal degree grows as $N^\alpha$ with $\alpha < 1$. It would be interesting to further explore quantum dynamics on such graphs, following [4, 7, 26].

The star graph has precisely the connectivity of the Dicke model of the superradiance transition [27]. Unlike our model, the Dicke model is expected to exhibit more conventional chaotic operator growth, with a finite Lyapunov exponent [28, 29, 30]. The crucial difference between the Dicke model and (7) is that the Hilbert space of the central site 1 is infinite dimensional in the Dicke model: it is a collective quantum harmonic oscillator (photon) mode. Truncating the central Hilbert space to be finite dimensional completely changes the quantum dynamics.

Similar to the Dicke model [31], we expect that models similar to (7) are amenable to experimental quantum simulation using trapped ions. As such experimental systems can exhibit $\sim 100$ quantum degrees of freedom, perhaps basic questions about bounds and universality in quantum chaos at finite temperature are better accessed in such experimental systems than in present day numerical simulations.

## ■ Acknowledgments

I thank Rahul Nandkishore, Xiao-Liang Qi, Ana Maria Rey and Koenraad Schalm for useful discussions. This work was supported in part by the Gordon and Betty Moore Foundation's EPiQS Initiative through Grant GBMF4302. This work was done in part at the "Chaos and Order" workshop at the Kavli Institute for Theoretical Physics, which is supported in part by the National Science Foundation under Grant No. PHY-1748958.

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
