# Peer review of "Quantum many-body dynamics on the star graph"

_SciPost Physics_

## Round 2 · Referee Report · Anonymous · 2019-6-16

Strengths

1) This paper presents some simple Hamiltonians with one and two spin interactions that show very fast growth of OTOCs, superficially faster than expected by suggested bounds on scrambling. This should help refine how bounds on scrambling should be stated in order to make them robust to special cases like the ones discussed here. The author proposes in the introduction to define the scrambling time as the time needed to produce near maximal entanglement. In this view the behavior of the OTOC is not directly addressing scrambling in all systems, although it may be a good diagnostic in some systems. The examples shown here illustrate cases where the OTOC is not a reliable measure of scrambling.

Weaknesses

1) The suggestion (e.g. 2nd sentence of abstract) of a phase transition should be stated differently. We know from studies of MBL that these questions are very challenging and one should not draw such conclusions just from numerical evidence like what is presented here. The situation is that the author finds that there is a crossover to a slower dynamics at lower T. Whether or not this should be classified as glassy behavior is not clear, nor whether this might be a true phase transition rather than a crossover.

2) There is another viewpoint of the system (eq 7) that is implicit here and might deserve more mention: The only interactions in (7) are 3 terms, each coupling one of the Pauli operators of the central spin to its particular linear combination of Pauli operators of the other spins. Thus this system has 3 very special operators at large N, which are these operators that are coupled to the central spin. These operators serve as the "bus" that makes these systems have very fast dynamics when viewed in terms of the single-spin Pauli operators. But is it just fast dynamics of operators spreading _on_ to the "bus"? Can one make a set of all the other operators that are orthogonal to these "bus" operators and ask if those other operators have more "normal" dynamics?

Report

This paper adds an interesting new example to the discussion of scrambling bounds and definitions. I think it adds enough that it should be published after the author considers the points above.

Requested changes

see under "weaknesses".

---

## Editorial Decision

editor-in-charge_assigned